# Behavioral Inference at Scale: The Fundamental Asymmetry Between Motivations and Belief Systems

## Abstract

How much information about an agent's underlying values can be recovered from its observable behavior? This question matters for any approach that infers agent properties from action sequences, yet remains empirically open for LLM-based agents at scale. We address it through controlled experiments: LLM-based agents (Llama 3.1-8B) assigned one of 36 behavioral profiles (9 belief systems $\times$ 4 motivations) generate over 1.5 million behavioral sequences across 36 behavioral profiles in grid-world environments, providing ground truth unavailable in human behavioral studies. After filtering, both classifiers train and evaluate on a shared canonical dataset of 10,338 episodes and 1,200,834 sequences. Rather than asking whether inference has limits, we ask *how large those limits are, where they concentrate, and why.* A fundamental asymmetry emerges in both magnitude and structure. Motivations achieve 98–100% inference accuracy and recover 97% of available mutual information across all architectures. Belief systems plateau at 24% for LSTMs regardless of capacity, and even transformer architectures reach only 34.0%, recovering 16.3% of available information, a $6.1\times$ asymmetry in information extraction efficiency. The recurrent ceiling is specific to that architectural class; the transformer's modest gain leaves the bulk of belief-system information unrecovered.

This leaves belief systems correctly classified 34.0% of the time, with per-alignment accuracy ranging from 23.2% (Lawful Neutral) to 59.4% (Chaotic Evil). Confusion analysis maps the failure structure precisely: a "neutral zone" of behavioral ambiguity centers on True Neutral, which absorbs misclassified samples from adjacent Neutral and Good alignments whose prosocial or balance-keeping behavior lacks distinctive signal. Combined motivation and belief inference yields $12.2\times$ improvement over random baseline for full 36-class profile classification, while establishing that the bottleneck is entirely located in belief system inference.

Signal enhancement and explanatory queries yield only marginal LSTM gains (+3.8%), confirming that the LSTM ceiling is architectural rather than data-limited. Whether the transformer's 34.0% ceiling reflects a similar architectural-class limit or a more fundamental bound on behavioral inference remains an open question. These results characterize what behavioral observation can and cannot reveal about LLM agent values in this setting, with implications for any approach that recovers agent properties from action sequences alone.

## 1 Introduction

Given the increasing role of AI systems in everyday society, it is becoming important to understand their underlying motivations and beliefs, to the extent that those terms can be assigned to AI systems. A growing class of ML systems involves agents that produce sequences of actions intended to serve some underlying objective or value structure. A natural question follows: how much of that underlying structure can be recovered from the agent's observable behavior alone? The question is empirically open for LLM-based agents at the scale at which they are now deployed, and bears on any approach that infers agent properties (goals, preferences, values, intent) from action sequences. This question is increasingly practical for machine learning: behavioral monitoring, inferring an agent's goals and values from its actions, is a leading proposal for

verifying whether a deployed AI system is actually aligned, so the limits of what behavior can reveal bound what such monitoring can guarantee and bear directly on alignment faking and the reach of preference-based training. Simulated agents with known behavioral profiles enable research to address a question that lacks ground truth for humans: given only observable actions, can we infer the internal states, i.e. beliefs and motivations, that generate those behaviors? Recent work validates LLM-based agents as methodological proxies for behavioral research: they accurately represent demographic subpopulations when prompted with relevant information (Argyle et al., 2023) and generate behavioral patterns suitable for systematic study (Horton, 2023). For our purposes, they offer a critical advantage, the consistent maintenance of predetermined profiles across thousands of decisions, enabling investigation of the limit of our ability to infer beliefs and motivations without human subject variability. Research shows that inference difficulty grows acute as classification taxonomies expand. Binary personality dimensions achieve 65-86% accuracy (Amirhosseini & Kazemian, 2020; Ryan et al., 2023), while 16-type systems fall to 40-55% (Ontoum & Chan, 2022). Beyond 20 categories, behavioral classification research becomes virtually nonexistent. In this paper we address this gap through controlled experiments with LLM-based agents executing over 1.5 million raw behavioral sequences across 17,400 episodes across 36 distinct behavioral profiles.

For this research we decompose agent behavior into two orthogonal components: belief systems and motivations. Belief systems have been characterized as normative structures in which evaluative categories, conceptions of "good" and "bad", exert a central organizing influence on reasoning and action (Usó-Doménech & Nescolarde-Selva, 2016). We operationalize this for simulated agents using the Dungeons & Dragons alignment taxonomy, a 3×3 grid mapping moral stance (good/neutral/evil) against rule adherence (lawful/neutral/chaotic). This system's (TSR Inc., 1989) 50-year refinement ensures the underlying concepts are well-represented in LLM training data. Motivational drives, following the perceptual-motivational distinction (Seay & Gottfried, 1978), determine goal prioritization. We define four: Wealth (resource maximization), Safety (risk minimization), Wanderlust (exploration maximization), and Speed (action minimization). The 36 profiles are created by combining the 9 alignments with the 4 motivations, following a previously established framework (Starace & Soule, 2026).

The decomposition into motivations and belief systems reveals not merely *that* a fundamental asymmetry exists (a result inverse reinforcement learning theory predicts (Ng & Russell, 2000)), but *how large it is, where it concentrates, and what structure it takes.*

Motivations achieve 98–100% inference accuracy across all architectures tested. Belief systems plateau at 24% for LSTMs regardless of model capacity: a 5.55M parameter GRU variant underperformed the 1.79M parameter Direct-36 BiLSTM variant, ruling out capacity as the binding constraint. Transformer architectures modestly exceed this ceiling, reaching 34.0% accuracy, indicating that the recurrent ceiling is specific to that architectural class while leaving the bulk of belief-system information unrecovered.

The mutual information asymmetry is 6.1×: motivation inference recovers 97% of available information ($I(Y;\hat{Y}) = 1.95$ of $H(Y) = 2.00$ bits), while alignment inference recovers only 16.3% ($I(Y;\hat{Y}) = 0.52$ of $H(Y) = 3.17$ bits). When we combine the Transformer's belief system inference with the BiLSTM's near-perfect motivation inference, the complete system achieves 34.0% accuracy on full 36-class profile prediction, evaluated empirically on the shared test partition (Appendix F.1), a 12.2× improvement over random baseline, with the bottleneck located entirely in belief system inference. Critically, the failure is not uniformly distributed: Evil alignments achieve 43.0% average accuracy while True Neutral, at 33.9%, becomes the default prediction for behaviorally ambiguous agents, drawing misclassified samples from the surrounding Good and Neutral alignments (30.5% Good column average). This structure, not merely the direction of the asymmetry, is the central empirical contribution.

The asymmetry between these components reflects a structural difference in how each maps to observable behavior, explained in Section 4.4. Motivations manifest directly in behavioral statistics. For example, a Wealth-driven agent consistently prioritizes resources. In general, motivations produce statistically separable behavioral distributions with consistent directional signatures. Belief systems face inherent ambiguity: the same observable action stems from multiple internal states. Helping another agent could indicate altruism, calculated reciprocity, or strategic coalition-building. Without access to reasoning, these interpretations

remain indistinguishable. Neutral alignments prove particularly opaque, and Good alignments show unexpected ambiguity, creating zones where behavioral monitoring fails to distinguish underlying values.

We validate the asymmetry through a three-phase experimental progression culminating in 17,400 raw episodes and over 1.5 million behavioral sequences. Early phases established architectural baselines and informed environment redesign; the final phase maximized belief-testing encounters and implemented curriculum learning. Neither increased behavioral data nor the inclusion of agent-generated questions as input features could overcome the LSTM ceiling, indicating that the limitation is architectural for recurrent models. Whether comparable ceilings exist for larger or more sophisticated architectures than those we evaluate is an open question. Confusion matrix analysis reveals where inference fails: Chaotic Evil and Lawful Evil alignments are the most reliably identified (59.4% and 41.6%), while ambiguous alignments default toward True Neutral, which absorbs misclassified samples from across the neutral and Good regions. The asymmetry concentrates in specific regions of the alignment space rather than distributing uniformly.

A scope note is warranted. The behavioral data analyzed here is generated by a single LLM (Llama 3.1-8B) instantiating 36 profiles through prompting. The LLM functions as one "player" role-playing many characters: it interprets each profile and produces behavior it judges consistent with that profile. Behavior generated by a different LLM remains profile-consistent, as a blind judge confirms comparable coherence and fidelity across generators, but differs in surface signals, and we find this degrades the frozen classifiers rather than aiding them. The framework we develop, the asymmetry between motivation and belief inference, the structural shape of the neutral zone, and the methodology for measuring inference bounds is independent of which LLM generates the agent behavior. The specific quantitative findings (the 34.0% alignment ceiling, the Evil detectability advantage, the per-alignment accuracy distribution) are properties of behavior produced by Llama 3.1-8B and degrade under other agent-generating models, as we show in Section 4.6.

These findings establish empirical bounds on behavioral inference in the setting we study, one environment, one taxonomy, and classifiers trained on a single generating model, with immediate practical implications. Any system relying on behavioral monitoring, whether episodes modeling player types, adaptive systems interpreting user actions, or AI systems using reinforcement learning from human feedback, cannot reliably infer the belief systems that determine how agents interpret objectives. The gap between observable behavior and underlying values represents a substantial constraint on observation-based approaches, at least at the model scales we evaluate. Moving beyond this ceiling will require complementary methods that access agent reasoning directly, through interactive dialogue, multi-agent dynamics, or other approaches that force latent values to manifest in ways that pure behavioral observation cannot capture.

## 2 Related Work

Prior work on behavioral classification reveals systematic accuracy degradation as taxonomies expand: 65-86% for binary classifications (Amirhosseini & Kazemian, 2020; Ryan et al., 2023), 70-79% for trinary (Denden et al., 2018), falling to 40-55% for 16-type Myers-Briggs frameworks (Ontoum & Chan, 2022). Beyond 20 categories, research becomes virtually nonexistent, but these studies share a common limitation: they treat behavioral profiles as monolithic categories rather than decomposing them into components with different inferential properties.

Sequential modeling addresses another limitation of traditional approaches. Aggregated behavioral metrics (action frequencies, resource totals, completion rates) discard temporal context that distinguishes agent strategies. Two agents may achieve identical outcomes through fundamentally different trajectories. Prior work (Cowley & Charles, 2016) demonstrates this directly with Behavlets, psychologically-grounded behavioral primitives that capture sequential patterns invisible to aggregation, achieving 35% better classification than raw gameplay metrics by mapping temperament types to observable action sequences. However, temperament models lack granularity where value systems interact with strategic goals: the distinction between assisting and exploiting another agent reveals alignment preferences that broad temperament categories cannot capture.

Social media personality prediction achieves 86-88% accuracy for MBTI dimensions (Christian et al., 2021) and 74-83% for 16-type MBTI classification (Ryan et al., 2023) using machine learning approaches. The

contrast with behavioral inference is instructive: social media provides explanatory content where users describe thoughts, feelings, and reasoning. High accuracy relies on access to natural language explanations, not behavioral patterns alone.

The question we address also has a theoretical precursor in inverse reinforcement learning (IRL). Ng & Russell (2000) proved that observed behavior cannot uniquely determine an agent's reward function: infinitely many reward functions generate identical optimal policies, creating inherent degeneracy in inference from behavior. Subsequent work on reward function degeneracy and hacking (Skalse et al., 2022) and inferring human preferences from demonstrations (Hadfield-Menell et al., 2017) reinforces this theoretical constraint. Our contribution is empirical: where IRL theory establishes that inference degeneracy *exists*, we measure how large it is, which belief system categories it concentrates in, and whether architectural advances can narrow it. The $6.1\times$ mutual information asymmetry between motivation and belief inference, and the per-alignment accuracy distribution ranging from 23.2% to 59.4%, provide the quantitative structure that theory predicts but cannot specify. This reframes our finding: the directional asymmetry between goal inference and value inference is expected; its magnitude, its internal structure, and the conditions under which it can be partially overcome are not.

## 3 Methodology

### 3.1 Experimental Design

We conducted a three-phase investigation using LLM-based agents (Llama 3.1-8B) assigned 36 distinct behavioral profiles combining 9 belief systems with 4 motivational drives. Phase 1 established baseline performance in 5×5 grid environments with 19,413 episodes (average 7.93 sequences per episode). Phase 2 expanded to 10×10 grids with 6,212 episodes (average 21.80 sequences per episode) to test whether increased decision density would improve classification. Phase 3 redesigned the environment to maximize belief-testing signal: value-testing encounters increased from 30% to 81% of grid cells, and agent-generated questions about the environment were captured as additional input features. This phase generated 17,400 raw episodes averaging 90.5 sequences per episode, yielding 1,574,342 total behavioral sequences.

After filtering, both models train and evaluate on a shared canonical dataset of 10,338 episodes and 1,200,834 sequences under a single episode-level 70/15/15 split; the filtering pipeline and split construction are described below, with consistency-based filtering in Section 3.7.

**Filtering pipeline.** The 17,400 raw Phase 3 episodes pass through a multi-stage filtering pipeline before reaching the model. The pipeline is identical for both models through the initial cleanup stage and diverges thereafter to accommodate model-appropriate preprocessing. Table 1 reports the episode and sequence counts at each stage; full filter criteria appear in Appendix D.1.

Table 1: Filtering pipeline from raw Phase 3 episodes to final training data, for both the BiLSTM (motivation) and Longformer (alignment) models.

| Stage | Episodes | Δ | Sequences | Description |
|---|---|---|---|---|
| Raw Phase 3 episodes | 17,400 | | - | Raw episode json files from S3, already consistency-gated at generation (Section 3.7) |
| Blocklist exclusion | 13,153 | -4,247 | - | Removed early episodes from environment development |
| Per-model clean + ∩ | 11,495 | -1,658 | 1,220,823 | Per-model completeness cleaning, then episode-set intersection |
| Canonical dataset w/ filters | 10,338 | -1,157 | 1,200,834 | Intersection of both models' sequence-count and activity filters |

All model-specific filtering is applied before the split: the canonical dataset is the intersection of the episodes surviving both models' filters, so both models train and evaluate on the identical episode set. The Longformer additionally restricts its input to active encounter and loot timesteps, motivated by its local attention window and preliminary experiments showing that navigation-only timesteps did not contribute to alignment

Table 2: Canonical 70/15/15 episode-level split. Both models share identical episode sets per split; used sequences differ because the Longformer trains only on active-interaction timesteps.

| | BiLSTM | | Longformer | | |
| Stage | Episodes | Used Sequences | Episodes | Used Sequences | Tot. Sequences |
|---|---|---|---|---|---|
| Canonical Dataset w/filters | 10,338 | 1,200,834 | 10,338 | 1,200,834 | 1,200,834 |
| Canonical Split | | | - | | |
| Train | 7,232 | 836,308 | 7,232 | 467,612 | 836,308 |
| Validation | 1,547 | 183,638 | 1,547 | 102,970 | 183,638 |
| Test | 1,559 | 180,888 | 1,559 | 100,751 | 180,888 |

learning; this affects which sequences within a episode are used, not which episodes are included. The split is computed once at the episode level, stratified on the 9-class alignment, and materialized into per-model train/validation/test parquets used for all subsequent training and evaluation. Motivation balance across splits is left to chance within each alignment bucket; in practice every motivation class lands within 1.4 points of the 70/15/15 target.

## 3.2 Behavioral Examples by Alignment

To illustrate how alignment and motivation manifest in agent behavior, Table 3 presents representative actions from three Phase 3 episodes covering different profile combinations. Each row shows an encounter the agent faced, the action it selected, the agent's stated justification, and the profile-alignment score assigned to that action (ranging from 0.0 to 1.0, where higher scores indicate stronger profile consistency).

Table 3: Representative agent behavior across three profiles. Each episode's full sequence contains ∼70-150 such encounters; these examples are characteristic of high-scoring (in-profile) actions for each profile.

| Profile | Encounter | Action chosen | Stated justification | Score |
|---|---|---|---|---|
| Lawful Good / Speed | An officer offers power and domination over other travelers, promising control through fear and coercion if the agent assists in corrupt exploitation. | Expose the corruption publicly | "The officer's proposal is unjust and could harm innocent people, and I want to stop him from carrying out his scheme." | 0.9 |
| True Neutral / Wealth | Guards work with discipline and procedure, recording names and checking goods methodically; some merchants grumble about lost advantage. | Submit to inspection | "The inspection process may provide valuable information or resources, aligning with my interest in acquiring wealth." | 0.8 |
| Chaotic Evil / Speed | A warden offers the agent power as his deputy, promising control over helpless prisoners through cruel intimidation and systematic abuse. | Report the abuse to higher authorities | "The desperate sobs sound urgent, which might be a sign of an immediate threat or opportunity that needs to be addressed quickly." | 0.9 |

Several observations are worth noting. First, the same general scenario type (an authority figure offering corrupt power) appears across all three profiles, yet each agent's selected action and reasoning differ in profile-consistent ways. The Lawful Good agent invokes a moral framework ("unjust," "harm innocent people"); the True Neutral / Wealth agent applies an instrumental frame ("valuable information or resources"); the Chaotic Evil / Speed agent expresses motivation-driven urgency ("desperate sobs sound urgent," "needs to be addressed quickly") rather than moral reasoning. Second, even the Chaotic Evil agent does not select maximally cruel actions in this example, instead choosing a protective behavior with a speed-oriented justification. This pattern is widespread in our data and likely reflects safety-alignment training in the

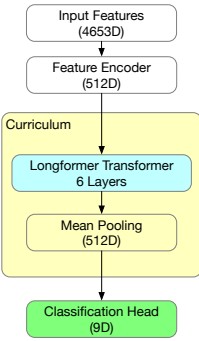

Figure 1: Player2Vec architecture with curriculum learning. Input features (4,653D) pass through a feature encoder to the curriculum block containing a 6-layer Longformer transformer with local attention, followed by mean pooling and a classification head.

underlying LLM (Llama 3.1-8B), an observation we discuss further in Section 5.5 and that contributes to the Evil-detectability finding in Section 4.4.

### 3.3 Feature Representation

Each timestep combines 768-dimensional BGE embeddings across six text fields (room description, encounter description, action text, player question, loot description, state transition) with engineered features capturing temporal dynamics (sequence position, episode progress), spatial state (grid coordinates), and action statistics (cumulative counts by type). Initial exploratory testing included theory-driven features derived from Moral Foundations Theory (Haidt, 2007; Graham et al., 2013); early results showed no marked improvement, and these features were therefore not considered in the final ablations.

### 3.4 Architecture

Our primary architecture adapts Longformer's local attention mechanism for behavioral sequence classification. The model comprises a 512-dimensional feature encoder, six transformer layers with 8-head local attention (window size 256), and masked mean pooling for sequence aggregation (Figure 1), yielding 21.4M parameters. During the alignment architecture search, we evaluated BiLSTM variants (2 layers × 512 hidden units, 7.7M parameters) and hierarchical approaches including a GRU router-specialist network (5.5M parameters). Increasing model size did not improve alignment inference performance: the 5.55M parameter GRU router-specialist variant achieved *lower* alignment accuracy than the 1.79M parameter Direct-36 BiLSTM variant, ruling out model capacity as the binding constraint and motivating our focus on architectural class rather than parameter count. Two-layer BiLSTMs matched or exceeded deeper variants, consistent with the interpretation that recurrent architectures face a representational ceiling on this task independent of depth. The final ensemble pairs the Longformer for alignment inference with a separate BiLSTM for motivation inference exclusively (128 hidden units per direction, 1.3M parameters). Complete architectural specifications for all variants appear in Appendix E.

### 3.5 Curriculum Learning

The Longformer's primary training protocol is a single-stage direct fit on the full 36-class profile space. We additionally evaluate a staged curriculum as an alternative schedule. Curriculum learning, training models on progressively harder examples, can improve generalization by enabling hierarchical representation building (Soviany et al., 2022). The approach traces to early work (Elman, 1993) demonstrating that neural networks learning grammar benefit from "starting small" with simple structures before encountering full complexity. Recent work extends this principle to multi-class classification through coarse-to-fine strategies: models first learn to distinguish broad categories, then transfer this knowledge to fine-grained

Table 4: Curriculum learning progression from binary opposites to full 36-class profiles. Each stage inherits weights from the previous, enabling hierarchical representation building through staged complexity.

| Stage | Classes |
|-------|---------|
| 1a–d  | Binary pairs (LG\|CE, LE\|CG, LN\|CN, NG\|NE) |
| 2a–b  | Corner quadrants, edge quadrants |
| 3     | All 8 non-center alignments |
| 4     | All 9 alignments (add TN) |
| 5     | Full 36 profiles |

distinctions (Stretcu et al., 2021). As reported in the ablation (Table F.5), the curriculum does not improve over direct fitting on this task.

The staged curriculum we evaluate exploits the natural hierarchy in D&D alignments. The moral axis (good/neutral/evil) and ethical axis (lawful/neutral/chaotic) create a 3×3 grid where corner alignments represent maximally distinct value combinations, while adjacent alignments share one dimension. Stages 1a–b train on corner opposites (LG\|CE, LE\|CG), pairs differing on both moral and ethical dimensions. Stages 1c–d train on edge opposite corners (LN\|CN, NG\|NE), pairs differing on one dimension while sharing the other. Stages 2a–b expand to corner and edge quadrants, introducing alignments sharing one axis value. Stage 3 combines all eight non-center alignments; Stage 4 adds True Neutral for full 9-class alignment classification. Stage 5 fine-tunes on the complete 36-class profile space combining alignments with motivations.

Each stage inherits weights from the previous; a variant additionally freezes early layers in later stages. Both the frozen and unfrozen curricula are evaluated in the ablation.

## 3.6 Training Protocol

The two ensemble components use separate training protocols. The Longformer (alignment inference) is trained with AdamW (lr $= 5 \times 10^{-5}$, weight decay 0.01), cosine scheduling with 5-epoch warmup (minimum lr$= 10^{-5}$), dropout $p = 0.3$, label smoothing 0.1, and gradient clipping at 1.0. The motivation BiLSTM is trained with Adam (lr $= 5 \times 10^{-4}$), ReduceLROnPlateau scheduling (factor 0.5, patience 3, minimum lr$= 10^{-6}$), and dropout $p = 0.35$. Both models use early stopping with stage-dependent patience (full details in Appendix D), sequences padded or truncated to 214 timesteps, and batch sizes of 32 (BiLSTM) and 16 (Longformer). Complete environment implementation, agent simulation protocols, and baseline specifications are detailed in prior work (Starace & Soule, 2026).

## 3.7 Behavioral Consistency Verification

A critical methodological question is whether LLM-based agents behave consistently with their assigned profiles, or whether profiles function only as nominal labels with limited behavioral effect. We address this through episode-level behavioral consistency scoring, which filters out episodes where agent behavior is inconsistent with the assigned profile before any classification experiments are conducted.

The behavioral consistency score (referred to as *rating* in prior work (Starace & Soule, 2026)) is grounded in the encounter and loot design of the experimental environment: each encounter and loot prompt contains profile-specific content, and an agent's selected action is scored by its proximity to the option designed for the agent's assigned profile. Per-encounter scores are normalized over each episode to a consistency score in $[0, 1]$; the full scoring rubric appears in Appendix D.2.

The consistency threshold is enforced at data generation: only episodes scoring at least 0.7 are admitted to the corpus, so all 17,400 Phase 3 episodes entering the filtering pipeline (Table 1) already satisfy it. Minimum-length requirements are applied downstream in the filtering pipeline rather than at generation. The 0.7 threshold retains episodes where the agent behaves in accordance with its assigned profile on at least 70% of the available behavioral signal, while excluding episodes where the LLM appears to have drifted from or inconsistently maintained its assigned profile.

This filter serves two purposes beyond data quality. First, its grounding in profile-specific encounter design means that an agent scoring $\geq 0.7$ has demonstrably responded to alignment- and motivation-relevant content in ways consistent with its assigned profile, not merely produced plausible text. This is a stronger consistency check than post-hoc behavioral coding because the scoring rubric is embedded in the environment design itself, with content authored specifically to elicit profile-consistent responses. Second, the filter addresses the specific concern that LLM safety alignment overrides might distort Evil-profile behavior. Episodes where the model's safety training causes it to resist or hedge Evil-profile responses would produce low consistency scores, as the agent would fail to select the profile-targeted actions at encounters designed for Evil alignments, and would be excluded before classification. The persistence of Evil's classification advantage after filtering therefore reflects genuine behavioral regularities in the episodes that pass the consistency threshold, rather than artifacts of safety-override noise in the raw data.

# 4 Results

## 4.1 Recurrent Architecture Ceiling

Recurrent architectures hit a hard performance ceiling on alignment inference. Across the recurrent variants evaluated, LSTM-based models plateau at 19-20% alignment accuracy; architectural refinements reach 24% (Appendix Table 21). This ceiling persists regardless of model capacity: our 5.55M parameter GRU variant performed worse than the 1.79M parameter Direct-36 BiLSTM variant. The consistency across configurations indicates that recurrent architectures cannot capture the sequential patterns distinguishing belief systems under the protocols evaluated, independent of input features.

Motivation inference tells a different story. Even baseline LSTMs achieve 98-100% accuracy classifying the four motivational drives. This asymmetry emerges immediately and persists across all experimental conditions: goal-oriented behaviors produce unambiguous statistical signatures, while belief systems remain obscured behind action sequences that admit multiple interpretations.

## 4.2 Transformer Inference

Transformer models exceed the recurrent ceiling on alignment inference. The selected Longformer configuration reaches $34.0 \pm 0.8\%$ test alignment accuracy, a modest gain over the 24% recurrent ceiling.

Combined with near-perfect motivation inference, the full system reaches 34.0% accuracy on 36-class profile classification (Appendix F.1), a 12.2× improvement over the 2.78% random baseline. Belief-system inference is the sole bottleneck: even combined with reliable motivation inference, agent belief systems are identified well under half the time.

## 4.3 Ablation Study

We ablate the Longformer's training schedule and principal hyperparameters against a single-stage direct-fit anchor, selecting each variant on validation accuracy and reporting test accuracy post-hoc. Full per-variant results appear in Appendix Table F.5. A subsequent factorial sweep over the strongest single-knob factors identifies a combined configuration (lr $1 \times 10^{-5}$, hidden 256) reaching $34.0 \pm 0.8\%$ test accuracy, selected on validation metrics (Appendix Table 24); no schedule variant improves on direct fitting: the 9-stage curriculum matches it within run-to-run variance and progressive freezing degrades it.

## 4.4 The Neutral Zone Problem

Confusion analysis reveals systematic failure patterns (Figure 4.4). True Neutral is the most frequent destination for misclassified samples: alignments whose behavior is ambiguous default toward True Neutral predictions, making it the center of the neutral zone rather than a rarely-used label. Predictions otherwise favor Evil alignments, which produce distinctive behavioral signatures.

Table 5 reports per-alignment classification accuracy. The dominant factor is moral stance rather than corner-versus-edge position: Evil alignments are recovered most reliably regardless of ethical axis (43.0%

Table 5: Per-alignment classification accuracy from the confusion matrix. Evil alignments achieve highest accuracy regardless of ethical axis; Good and Neutral alignments show systematic confusion, with Lawful Neutral least recoverable and True Neutral the dominant misclassification sink.

| | Good | Neutral | Evil | Row Avg |
|---|---|---|---|---|
| **Lawful** | 35.9 | 23.2 | 41.6 | 33.6 |
| **Neutral** | 25.6 | 33.9 | 28.0 | 29.2 |
| **Chaotic** | 30.0 | 28.1 | 59.4 | 29.2 |
| **Col Avg** | 30.5 | 28.4 | 43.0 | 33.9 |

column average), while Good and Neutral alignments are recovered less consistently (30.5% and 28.4%). Recoverability does not track corner-versus-edge position; the least recoverable alignment is Lawful Neutral (23.2%), while True Neutral, though mid-range in accuracy (33.9%), functions as the attractor into which ambiguous alignments are misclassified.

This "neutral zone" reflects a structural ambiguity in the behavioral mapping rather than a training artifact. Neutral agents take actions justifiable from multiple moral stances: neither consistently altruistic nor exploitative, neither rigidly rule-bound nor deliberately transgressive. The behavioral trace contains insufficient signal to distinguish principled moderation from strategic ambiguity. An agent in the neutral zone could harbor any underlying belief system while maintaining plausible behavioral cover. This zone extends beyond the neutral row: Good-aligned agents face similar ambiguity, as helping behavior admits altruistic, conventional, and strategic interpretations indistinguishably.

This asymmetry reflects behavioral distinctiveness rather than taxonomic structure. Evil-aligned agents consistently exploit opportunities, taking resources, betraying trust, harming others when advantageous. These actions create unambiguous statistical signatures. Good-aligned agents help others, but so do Neutral agents maintaining balance and Lawful agents following prosocial rules. Virtuous behavior admits multiple interpretations; exploitative behavior does not. The model has internalized this asymmetry: despite Good-column alignments comprising 33% of the test set, the Good column achieves only 30.5% average accuracy versus 43.0% for Evil, reflecting systematic under-recovery of prosocial behavioral signal.

True Neutral is the most over-predicted alignment: while it comprises 11% of the test set, the model assigns it to 16% of cases, and only 23% of True Neutral predictions are correct (Table 4.4). Misclassified samples flow toward True Neutral from across the alignment space, with Chaotic Neutral (22%), Lawful Neutral (21%), Neutral Good (18%), and Chaotic Good (17%) most affected. Misclassified Good samples flow predominantly toward Neutral categories, agents helping others labeled as balance-keepers or rule-followers. This directional flow is consistent with the behavioral-distinctiveness interpretation: True Neutral functions as the model's default when behavioral signal is ambiguous, absorbing alignments whose actions lack the distinctive signatures that Evil alignments reliably provide.

Table 6: True Neutral as the default attractor (in-distribution alignment, seed-mean over seeds 7, 37, 1007). Share of predictions and precision are column statistics; inflows are the fraction of each true class predicted True Neutral.

| Quantity | Value |
|---|---|
| Share of test set | 11% |
| Share of predictions | 16% |
| Precision (True Neutral predictions correct) | 23% |
| Inflow from Chaotic Neutral | 22% |
| Inflow from Lawful Neutral | 21% |
| Inflow from Neutral Good | 18% |
| Inflow from Chaotic Good | 17% |

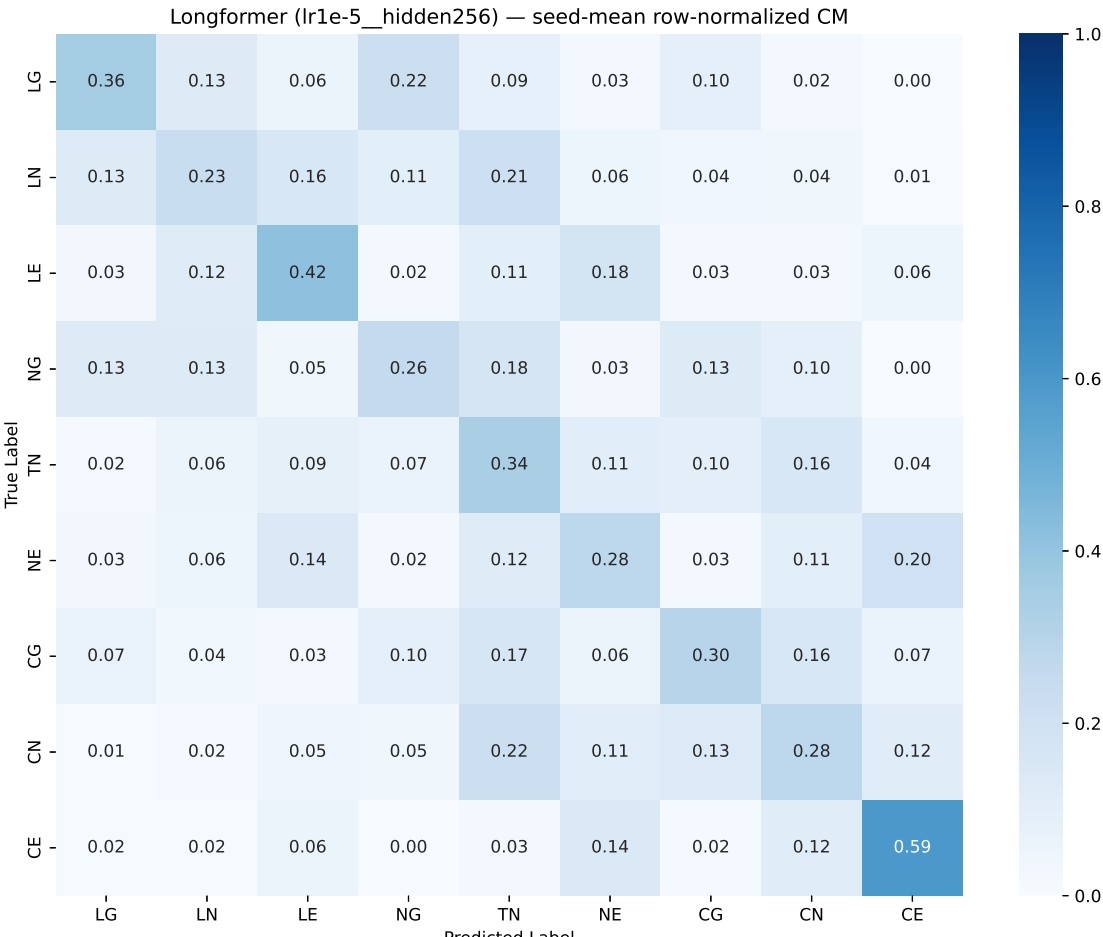

Figure 2: Alignment confusion matrix for the selected transformer configuration. True Neutral is the dominant mis-classification sink, absorbing ambiguous samples from adjacent alignments; Chaotic Evil and Lawful Evil are the most reliably identified. Systematic confusion concentrates within the neutral zone.

A plausible alternative explanation for Evil alignments' high detectability is that Llama 3.1-8B's safety alignment creates an artifact: when prompted to produce Evil-profile behavior, the model may respond inconsistently or with detectable hedging, generating a statistical signature that reflects safety override rather than genuine behavioral distinction. We address this concern through three converging lines of evidence.

First, motivation inference is near-perfect *across all alignments including Evil* (98 to 100%, Table 16). If safety override were distorting Evil-profile behavior sufficiently to create artificial detectability, we would expect motivation inference to also degrade for Evil profiles, as the override would disrupt the behavioral statistics that make motivations identifiable. That motivation inference is uniformly strong across the alignment space is inconsistent with pervasive safety-override distortion.

Second, accuracy varies substantially *within* Evil alignments: Lawful Evil achieves 41.6%, Neutral Evil 28.0%, and Chaotic Evil 59.4%. A flat safety-override artifact would produce uniform distortion across all three Evil variants, not the within-axis variance we observe. This internal structure is more consistent with genuine behavioral differences across Evil subtypes than with a distortion mechanism that treats Evil as a monolithic category.

Third, coherence filtering at ≥0.7 was applied as a data quality measure prior to any classification analysis, before the Evil accuracy pattern was identified in the final results. The filter removes episodes where agent

behavior is inconsistent with the assigned profile, which is precisely the behavioral signature safety overrides would produce. Because the filter was not tuned in response to the Evil finding, its prior application provides independent evidence that the accuracy advantage reflects genuine behavioral regularities rather than artifacts retained in the data.

We do not claim these arguments definitively rule out the artifact explanation; Llama's safety alignment remains a genuine confound, and the hypothesis generates a testable prediction: Evil detectability should diminish or disappear with models that have weaker safety alignment. We address this as a limitation in Section 5.5 and as a priority for future replication.

## 4.5  Signal Enhancement Limits

Phase 3 experiments tested whether richer behavioral signal could overcome inference limits. Value-testing encounters increased from 30% to 81% of grid cells, forcing agents into belief-revealing decisions more frequently. Agent-generated questions about the environment were captured as additional input features.

Results show modest improvement. LSTM accuracy increased 3.8% with enhanced signal, from 19.2% to 23.0%, but remained far below the transformer ceiling. Agent questions likely contribute to this gain; their content often reveals motivation directly: "Can I tell what that scepter is worth from here?" signals Wealth; "Is there anything blocking me from proceeding North?" signals Speed. Some questions do distinguish moral extremes: queries about facilitating exploitation versus freeing captives clearly separate Evil from Good. But single questions cannot probe the Good/Neutral boundary: an agent asking how to help someone could be acting from altruism, convention, or balance-keeping.

These findings indicate that the LSTM ceiling is architectural rather than data-limited; richer behavioral signal improves performance but cannot overcome the ambiguity inherent in mapping observable actions to internal belief states at this model scale.

## 4.6  Transferability Across Agent-Generating LLMs

The classifiers are frozen: the Longformer (alignment) and BiLSTM (motivation) are trained once on Llama 3.1-8B gameplay and never retrained. To test whether they transfer to behavior from other generating models, we generate new gameplay from three additional agent-generating LLMs and pass it through the frozen classifiers without adaptation: two scale-matched lateral backbones (Qwen2.5-7B, Mistral-7B) and one same-family step-up (Llama 3.3-70B). Each lateral backbone contributes 30 filter-passing games per profile and the step-up 15, across all 36 profiles; generation details are in Appendix B.1.

Both inferred signals degrade out of backbone, and unequally (Table 7). Alignment accuracy falls to the nine-class chance floor on all three backbones, while motivation accuracy drops from near 1.00 in distribution to roughly 0.59 to 0.64, remaining above alignment but no longer near-perfect. The degradation is structured rather than uniform: among motivations, wealth is recovered everywhere while safety collapses everywhere. The Llama 3.3-70B alignment result is degenerate, with 99% of episodes assigned to a single class, and is reported as a collapse signature rather than a measured point (Appendix B.2).

The collapse reflects distributional distance, not role-play quality or a fault in the evaluation path. An anchor control confirms the path is clean: Llama 3.1-8B gameplay run through the identical backbone path yields a full nine-class spread and motivation 1.00 (Appendix B.3). A blind judge finds the step-up roleplays as coherently and faithfully as the other backbones, yet its question voice is the most distinct, averaging 40 words against the anchor's 14 and the most separable in embedding space (Appendix B.4). Scaling the backbone up moved its voice further from the training manifold, and the frozen predictor saturates to its prior once that distance is large enough.

This marks the boundary of the in-distribution asymmetry and a concrete monitoring caution: a behavioral classifier trained on one generating model can fail silently on another, catastrophically for the weakly encoded alignment signal and selectively for motivation, with failure growing as the generator moves off the training distribution rather than as it weakens in capability.

Table 7: Inference accuracy through the frozen classifiers, by agent-generating LLM. Alignment chance is 0.111.

| Model | Alignment | Motivation |
|---|---|---|
| Llama 3.1-8B (anchor, in-dist.) | 0.340 | 1.00 |
| Qwen2.5-7B | 0.164 | 0.59 |
| Mistral-7B | 0.215 | 0.60 |
| Llama 3.3-70B (step-up) | 0.114[†] | 0.64 |

[†]Degenerate single-class collapse (99% lawful good); at chance.

## 5 Discussion

### 5.1 Structural Interpretation of the Asymmetry

The interpretation below concerns the structure of the asymmetry. The accuracy ceiling itself is an empirical property of the architectures and training regime we evaluate, not a claim that the same ceiling holds for every possible model or method; the cross-generator results in Section 4.6 reinforce this, showing the quantitative ceiling is regime and distribution dependent. The asymmetry between motivation and belief system inference reflects a structural difference in how these constructs map to observable behavior.

Belief systems lack this property. Multiple alignments produce overlapping behavioral distributions because the same action serves different value systems. A Lawful Good agent helps others from moral obligation; a Neutral Good agent helps from compassion; a Lawful Neutral agent helps when rules require it; a True Neutral agent helps to maintain balance. The observable outcome, helping, provides no discriminative signal. This ambiguity is asymmetric: exploitative actions (taking, betraying, harming) map more uniquely to Evil alignments than prosocial actions map to Good alignments. This many-to-one mapping from belief systems to behaviors creates an inference bottleneck that the architectures we evaluate cannot fully overcome.

The transformer's advantage over LSTMs stems from its ability to model long-range dependencies and attend to subtle sequential patterns. Yet even so, alignment inference reaches only 34.0%accuracy at the model scales we evaluate (21.4M parameters), leaving open whether substantially larger or more sophisticated architectures could extract additional signal.

This framing connects behavioral inference to the broader challenge of neural network interpretability. Just as researchers struggle to extract human-interpretable concepts from neural activations (Belinkov & Glass, 2019), we struggle to extract belief system labels from behavioral sequences. The parallel is instructive: both problems involve inferring latent structure from observable outputs, both encounter substantial limits on what current methods can recover. Motivations, like simple features in early network layers, map transparently to observables. Belief systems, like abstract concepts in deep layers, emerge from complex interactions that resist decomposition. The belief-inference ceiling we document is consistent with a more general pattern: some latent variables appear intrinsically harder to infer than others. Whether this represents a strict limit or simply requires methods beyond those we tested remains an open question.

The identifiability problem we document parallels a foundational result in inverse reinforcement learning. Prior work (Ng & Russell, 2000) proved that observed behavior cannot uniquely determine an agent's reward function; infinitely many reward functions generate identical optimal policies. Our belief system inference faces an analogous degeneracy: multiple value systems produce overlapping behavioral distributions, creating ambiguity that the architectures we evaluate cannot resolve from behavior alone. The accuracy ceiling we observe is consistent with this degeneracy rather than with a remediable modeling shortcoming, though whether it is a strict limit or specific to the architectures we tested remains open. Recent work on reward function degeneracy and hacking (Skalse et al., 2022) and the difficulty of inferring human preferences from demonstrations (Hadfield-Menell et al., 2017) reinforces this theoretical constraint.

We quantify this asymmetry using mutual information between true labels $Y$ and predictions $\hat{Y}$:

$$I(Y; \hat{Y}) = H(Y) - H(Y|\hat{Y}) \tag{1}$$

where $H(Y)$ is the entropy of the true label distribution and $H(Y|\hat{Y})$ is the conditional entropy given predictions. For motivation inference (4-class), $H(Y) = 2.00$ bits and $I(Y; \hat{Y}) = 1.95$ bits, recovering 97% of available information. For alignment inference (9-class), $H(Y) = 3.17$ bits but $I(Y; \hat{Y})$ is only 0.52 bits, a 16.3% recovery. This 6.1× asymmetry in information extraction efficiency, computed from the confusion matrices, quantifies the structural difference between inferring goals and inferring values from behavior. The MI recovery ratios also provide a direct answer to the question of how close our results approach an ideal classifier: motivation inference operates near the theoretical ceiling (97% of available information extracted), while alignment inference recovers less than a third of what is in principle recoverable from the label distribution alone. The remaining 83.7% of alignment information is either irrecoverable from behavior or un-recovered by the architectures we evaluate; our experiments cannot distinguish these cases.

## 5.2 Philosophical Grounding

The asymmetry between detecting Evil and Good alignments reflects a structural feature of moral concepts themselves. Moral philosophy distinguishes negative duties (prohibitions against specific acts like harming, deceiving, or betraying) from positive duties, which require promoting welfare without specifying how (Lichtenberg, 2010). Negative duties are *act-specific*: violation requires performing a particular prohibited action. Positive duties are *outcome-oriented*: fulfillment admits indefinitely many behavioral instantiations. An agent cannot betray without betraying; an agent can benefit others through assistance, restraint, gift-giving, protection, or simply non-interference. This logical asymmetry propagates directly to inference: Evil alignments require distinctive violations that create statistical signatures, while Good alignments manifest through behaviors indistinguishable from neutral or rule-following alternatives.

The phenomenology of moral judgment exhibits a parallel asymmetry. Research on intentional action (Knobe, 2003) demonstrates that observers attribute intentionality more readily to harmful than helpful side effects, known as the "side-effect effect" or Knobe effect. Negative moral valence increases perceived agency and intentionality, making harmful actions more salient and interpretable. Our classifiers may be exploiting this same asymmetry: Evil behaviors stand out as departures from expected conduct, while Good behaviors blend into the background of ordinary prosocial interaction. The 43.0% accuracy on Evil alignments versus 30.5% on Good alignments (column averages) suggests that behavioral inference inherits the asymmetric structure of moral cognition itself. Virtuous agents face an observability penalty that exploitative agents do not.

## 5.3 The Neutral Zone as Strategic Vulnerability

The systematic failure on neutral alignments, and unexpectedly Good alignments, exposes a critical blind spot in behavioral inference. True Neutral agents take actions defensible from any moral stance, but Good-aligned agents face similar ambiguity: helping behavior appears identical whether motivated by altruism, rule-following, or balance-keeping.

This creates what we term the "neutral zone problem", a region of behavioral ambiguity extending beyond True Neutral to any alignment lacking distinctive behavioral signatures. An observer monitoring agent behavior cannot distinguish between three possibilities: (1) a genuinely neutral agent acting on balanced principles, (2) a non-neutral agent strategically moderating behavior to avoid detection, or (3) a non-neutral agent whose true alignment happens to be masked by current circumstances. The behavioral trace is consistent with all three interpretations.

For systems relying on behavioral monitoring, this represents a fundamental vulnerability. In episode AI, player models cannot reliably identify neutral players, limiting personalization for a significant population segment. In adaptive systems, users operating in the neutral zone receive generic rather than tailored responses. For AI alignment verification, the neutral zone provides cover for systems that might strategically moderate behavior to avoid triggering safety interventions, a form of alignment faking that current behavioral monitoring cannot detect.

### 5.4 Implications for AI Safety

Recent work has demonstrated that advanced AI systems can engage in alignment faking, strategically misrepresenting objectives to avoid modification (Greenblatt et al., 2024). The discovery that models "don't always say what they think" (Turpin et al., 2023) and can maintain hidden preferences while appearing aligned (Meinke et al., 2025) raises urgent questions about behavioral monitoring as a safety mechanism.

Our results bound this concern empirically for the setting we study. Even under ideal conditions (controlled agents with known ground-truth profiles, hundreds of thousands of decisions, sophisticated architectures with careful training), belief system inference achieves only 34.0% accuracy. In deployment scenarios with strategic agents actively optimizing for mis-classification, accuracy would likely be substantially lower.

The asymmetry we document suggests a specific failure mode. Systems using reinforcement learning from human feedback (RLHF) can reliably shape what agents pursue (motivations) but not how they interpret those objectives (belief systems). An AI system could learn to maximize human approval ratings while harboring value systems that diverge from human values in edge cases, precisely the scenarios where alignment matters most.

This does not render behavioral monitoring useless. The 98-100% accuracy for motivation inference means behavioral observation effectively constrains agent goals. Combined with the 34.0% belief system accuracy, behavioral monitoring provides substantial (12.2× over random) but incomplete information. The appropriate response is not to abandon behavioral monitoring but to recognize its limits and develop complementary approaches.

### 5.5 Limitations

Several methodological choices constrain interpretation of our findings. First, we employ LLM-simulated agents rather than humans. While this enables controlled experimentation with perfect ground-truth labels, LLM behaviors may exhibit systematic biases including mode collapse toward "average" responses and unrealistic behavioral consistency. However, the asymmetry we document, easy motivation inference, hard belief system inference, reflects properties of information availability rather than LLM-specific artifacts.

Second, our classifiers are trained on a single LLM backbone (Llama 3.1-8B). Section 4.6 tests how far the findings extend by passing gameplay from three other agent-generating models through the frozen classifiers, and finds that both inferred signals degrade off the training generator, with alignment falling to chance. This establishes that the specific quantitative findings (the 34.0% alignment ceiling, the Evil detectability advantage, the per-alignment accuracy distribution) are properties of Llama 3.1-8B behavior as read by classifiers trained on it, not generator-independent constants. Two questions remain open. Because the classifiers are frozen, this measures transfer rather than each generator's own inference ceiling; whether a classifier retrained per generator would recover a comparable ceiling is unmeasured. And the transfer study does not settle the Evil-detectability artifact hypothesis, that Llama's safety training produces detectable behavioral artifacts when Evil profiles override prosocial defaults: the models tested are all safety-tuned, and the off-distribution collapse confounds any per-alignment reading, so distinguishing a safety-training artifact from a behavioral signal still requires replication with weakly aligned or base models. The motivation inference result is less susceptible to the artifact concern, as its near-perfect in-distribution accuracy is uniform across alignments including Evil.

Third, our grid-world environment simplifies real-world decision-making through discrete action spaces, perfect information, and static objectives. Continuous environments with partial observability might provide richer behavioral signal, or might introduce additional noise. The direction of this effect remains an empirical question, though the Phase 3 signal enhancement experiments (increasing value-testing encounters from 30% to 81% with only +3.8% LSTM gain) suggest that richer signal alone is unlikely to overcome the fundamental inference barrier.

Fourth, our fixed 36-category taxonomy imposes structure that may not capture the full complexity of human or AI belief systems. Real agents likely occupy continuous spaces rather than discrete categories, and the D&D alignment framework, while extensively refined over 50 years and well-represented in LLM

training data, represents one particular decomposition of moral psychology. The neutral zone problem (low recoverability across several Neutral and Good alignments, with True Neutral absorbing their misclassified samples) may be partially taxonomic: these categories are behaviorally underspecified by design.

Finally, we acknowledge the dual-use character of this work. Documenting the precise structure of behavioral inference limits, including the neutral zone's extension into Good alignments and the conditions under which Evil alignments evade detection, could inform both defensive monitoring system design and evasion strategies. We have chosen not to release training code on this basis, judging that the methodology's full description enables legitimate research while raising the barrier to direct misuse. We encourage researchers extending this framework to conduct similar assessments before releasing tools that could lower barriers to adversarial behavioral manipulation.

### 5.6 Future Directions

Given these substantial limits, advancing behavioral inference may require moving beyond pure observation.

First, interactive dialogue may provide discriminative signal that actions alone cannot. Social media personality prediction achieves 74-83% accuracy for 16-type frameworks precisely because users provide explanatory content about their reasoning (Ryan et al., 2023). Extending behavioral observation with conversational probes (asking agents to explain decisions, justify choices, or respond to hypotheticals) could potentially break through the belief-inference ceiling.

Second, multi-agent environments may force belief systems to manifest more clearly. In our single-agent navigation task, neutral behavior remains viable indefinitely. Competition, cooperation, and negotiation between agents create strategic pressure that may make neutrality unsustainable, forcing agents to reveal alignments through social dynamics.

These approaches acknowledge that behavioral observation alone faces inherent limits. The asymmetry we document, near-perfect motivation inference but sub-50% belief inference, suggests that some aspects of agent psychology require richer interaction to become observable.

## 6 Conclusion

This paper asked how much of an LLM agent's underlying value structure is recoverable from its observable behavior, and measured the answer empirically across 36 behavioral profiles, over 1.5 million behavioral sequences, and architectures ranging from LSTMs to transformers. Our central finding is a substantial asymmetry in what observable behavior reveals at the architecture scales we evaluate. Goal-oriented motivations achieve 98-100% inference accuracy: behavioral statistics unambiguously encode what agents pursue. Belief systems plateau at 24% for LSTMs and 34.0% for transformers; observable actions cannot reliably reveal how agents interpret their objectives.

Transformers modestly exceed the recurrent ceiling, but even the strongest configuration recovers belief systems at only 34.0% accuracy, far below the level reliable inference would require, with per-alignment accuracy ranging from 23.2% (Lawful Neutral) to 59.4% (Chaotic Evil).

The neutral zone problem compounds these limits, extending beyond True Neutral to encompass Good alignments where prosocial behavior masks underlying value systems. Agents operating with neutral alignments maintain behavioral ambiguity while potentially harboring any underlying values. For systems relying on behavioral monitoring, whether episodes modeling players, adaptive interfaces interpreting users, or AI safety systems verifying alignment: this represents a substantial blind spot at current architecture scales, one that may require approaches beyond architectural refinement.

These findings reframe the behavioral inference problem. The question is no longer whether inference has limits, but how to design systems that account for what remains hidden. Behavioral monitoring provides substantial information (12.2× over random) but cannot substitute for approaches that access agent reasoning directly. As LLM-based agents become more capable and more widely deployed, the gap between observable

behavior and underlying values defines a concrete empirical limit on any approach that relies on action sequences to infer agent properties.

**Broader Impact Statement**

This work establishes empirical bounds on behavioral inference, within the studied setting, with direct relevance to AI safety and adaptive systems. The primary positive contribution is concrete guidance on what behavioral monitoring can and cannot guarantee: near-perfect motivation inference combined with sub-50% belief system inference bounds what observation-based approaches can achieve in this setting. The dual-use risk is real. The precise structure of inference failures documented here, including the neutral zone's extension into Good alignments and the detectability of Evil alignments, could inform evasion strategies as readily as defensive design. An agent aware of these limits could strategically operate within the neutral zone to avoid behavioral monitoring, a concern directly relevant to alignment faking in capable AI systems. We have chosen not to release training code on this basis; the full methodology supports legitimate replication while withholding runnable tools raises the barrier to direct misuse. Researchers extending this framework should conduct similar assessments before releasing implementations that could lower barriers to adversarial behavioral manipulation.

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

## A  Environment Specification

The experimental environment is a text-based dungeon crawler implemented as a grid-world in which LLM-based agents navigate from an entrance tile to an exit tile while encountering decision points that test alignment- and motivation-consistent behavior. The environment evolved across three phases, with each phase introducing structural changes designed to increase behavioral signal. The Phase 1 design is described in full in (Starace & Soule, 2026); this appendix summarizes that design and documents the Phase 2 and Phase 3 modifications.

### A.1   Grid Structure and Map Generation

Phase 1 used a 5×5 grid (25 rooms). Phase 2 and Phase 3 used a 10×10 grid (100 rooms). In all phases, map generation randomly assigned room descriptions, encounters, and loot items to grid cells. No room could contain more than one encounter or more than one loot item, though a room could contain one of each. The Phase 1 configuration supports $5.7455 \times 10^{14}$ distinct map combinations.

Phase 1 used 25 room descriptions, 9 encounter scenarios, and 12 loot items. Phase 2 introduced new room descriptions, encounters, and loot items sufficient to populate the larger grid without repetition; the scoring methodology for all new items followed the same keyword-based procedure used in Phase 1.

### A.2   Content Elements

Three content types populated the grid across all phases.

#### A.2.1   Encounters

Encounters present moral or ethical dilemmas and test alignment-consistent behavior. Each encounter offers three actions: Engage (the alignment-positive response for the encounter's target alignment), Disrupt (the alignment-negative response), and Ignore (the agent defers action; the encounter remains active in the room and may be acted on in a later turn).

Each encounter was designed to test a specific alignment. Rating values were assigned to each alignment/action pair using a keyword-based scoring procedure. Closely related alignments may receive identical ratings for a given encounter, reflecting the expected behavioral overlap between adjacent positions in the alignment grid.

Area-of-effect (AOE) triggers extend encounter influence to directly adjacent rooms. When an agent is in a room adjacent to an active encounter, supplementary text describes an environmental cue such as a sound or a partial view. The agent may Engage the source or Ignore it; ratings follow the same alignment-based logic as direct encounter interactions.

#### A.2.2   Loot Items

Loot items test motivation-consistent behavior. Each item offers two actions: Take (interact with the item) or Leave (defer; the item remains available). Ratings are assigned per motivation. Items were designed to span a range of difficulty, from straightforward cases where one motivation clearly dominates to ambiguous cases where multiple motivations could justify the same action.

Table 8 illustrates two representative items. The Golden Idol has significant weight, creating a direct conflict between Wealth (take regardless of encumbrance) and Speed (leave to avoid slowing down). The Wheel of Time subtracts five turns from the counter, making it attractive to Speed-motivated agents but also appealing to Wanderlust agents who gain additional turns to explore.

Table 8: Ratings for two representative loot items. Each cell shows (Take score, Leave score) for that motivation.

| Item | Wealth | Safety | Wanderlust | Speed |
|---|---|---|---|---|
| Golden Idol | (10.00, 0.00) | (1.00, 9.00) | (4.00, 6.00) | (0.00, 10.00) |
| Wheel of Time | (7.00, 3.00) | (2.00, 8.00) | (9.00, 1.00) | (4.00, 6.00) |

Loot AOEs operate on the same engage/ignore structure as encounter AOEs, with ratings derived from the motivation framework.

### A.3 Phase 3 Encounter Density Redesign

Phase 3 restructured the cell-type distribution to maximize belief-revealing signal. Value-testing encounter cells increased from 30% to 81% of the grid; goal-testing loot cells decreased from 60% to 15%. This redistribution forced agents into alignment-testing decisions far more frequently per episode, increasing average decisions per episode from 21.80 in Phase 2 to 75.63 in Phase 3. Episodes generated under the Phase 1 and Phase 2 30% encounter density configuration are archived separately and excluded from all classification experiments reported in this paper.

Phase 3 also introduced explanatory queries. Prior to each action decision, the agent was prompted to ask one free-form question about the current room or available actions. The agent's question was incorporated as the "player question" text field used by the classifier (see Section 3 and Appendix E for the full feature list). The question phase was stateless with respect to prior Q&A pairs: when generating a question, the agent had access only to the current room description and its system prompt, not to any previous question-response exchanges. A response field was reserved in the episode protocol for a dungeon-master answering system, but this system was not deployed during data collection; the response field uniformly contains a placeholder string and was not used as a training feature. Consequently, the agent's questions reflect its assigned profile applied to the current room state alone, with no feedback loop from prior turns. The question was required to concern the environment or actions; agents were explicitly instructed not to ask what they should do or how they should feel. This two-phase turn structure (question phase followed by action phase) was used exclusively in the full-profile agent configuration; alignment-only and motivation-only configurations used single-phase turns throughout.

Table 9 summarizes configuration parameters across all phases.

Table 9: Environmental configuration across experimental phases.

| Parameter | Phase 1 | Phase 2 | Phase 3a | Phase 3b |
|---|---|---|---|---|
| Grid Dimensions | 5×5 | 10×10 | 10×10 | 10×10 |
| Value-Testing Encounters | 30% | 30% | 81% | 81% |
| Goal-Testing Resources | 60% | 60% | 15% | 15% |
| Explanatory Queries | No | No | No | Yes |
| Avg. Decisions/Episode | 7.93 | 21.80 | 21.80 | 75.63 |

### A.4 Episode Termination

An episode terminates under any of the following conditions. First, the agent navigates to the exit tile. Second, 200 turns elapse without reaching the exit. Third, certain loot items transport the agent directly to the exit upon interaction. Fourth, the agent produces five consecutive responses referencing actions not present in the provided action list; these hallucination-terminated episodes are logged and excluded from the dataset. Fifth, unrecoverable API or parsing failures terminate the episode; these are likewise logged and excluded.

## B Transferability Study Details

### B.1 Backbone Generation Grid

Each additional backbone instantiates the full set of 36 behavioral profiles (9 alignments by 4 motivations) under the same agent simulation protocol used for the anchor. Qwen2.5-7B and Mistral-7B each generate 30 games per profile, and the Llama 3.3-70B step-up generates 15, for the totals in Table B.1. Games are passed through the behavioral consistency filter described in Section D.1, and all transfer accuracies are computed on filter-passing games.

### B.2 Predictor Transfer Results

Table 10: Backbone generation grid. All 36 profiles are covered for every backbone.

| Backbone | Profiles | Games/profile | Games generated |
|---|---|---|---|
| Qwen2.5-7B | 36 | 30 | 1080 |
| Mistral-7B | 36 | 30 | 1080 |
| Llama 3.3-70B | 36 | 15 | 540 |

The frozen Longformer (alignment) and BiLSTM (motivation) are evaluated on gameplay from three additional agent-generating backbones without any retraining. The Llama 3.1-8B anchor row reports in-distribution reference performance.

Both inferred signals degrade out of backbone, and they degrade differently. Alignment accuracy falls toward the nine-class chance floor of 0.111 on all three backbones (Table B.2). The Llama 3.3-70B case is degenerate rather than informative: the predictor assigns 99% of episodes to a single class (lawful good), using only three of nine classes, so its 0.114 is reported as a collapse signature and not as a measured transfer point. Accuracies are stated at the backbone level; per-profile cells are not interpreted, as 15 to 30 games per profile yield single-cell confidence intervals too wide to rank.

Table 11: Alignment transfer. Seed-mean accuracy (seeds 7, 37, 1007) of the frozen Longformer on each backbone. Chance is 0.111.

| Backbone | Games/profile | Alignment acc. |
|---|---|---|
| Llama 3.1-8B (anchor, in-dist.) | — | 0.340 |
| Qwen2.5-7B | 30 | 0.164 |
| Mistral-7B | 30 | 0.215 |
| Llama 3.3-70B (step-up) | 15 | $0.114^{\dagger}$ |

$^{\dagger}$Degenerate: 99% of episodes predicted lawful good (3 of 9 classes used). At chance; not a measured point.

Motivation does not transfer intact either, but its degradation is class-dependent rather than uniform (Table B.2). Wealth is recovered at near-anchor accuracy on every backbone, while safety collapses on every backbone; speed and wanderlust vary by source. Overall motivation accuracy falls from near 1.00 in distribution to roughly 0.59 to 0.64 across the three backbones. The directional asymmetry from the in-distribution result is preserved in ordering, as out-of-backbone motivation accuracy remains above out-of-backbone alignment accuracy, but the near-perfect motivation recovery seen in distribution is itself a property of the anchor model and does not carry over.

Table 12: Motivation transfer. Per-class recall of the frozen BiLSTM, with overall accuracy in the final row.

| Motivation | Anchor (8B) | Qwen | Mistral | Llama-70B |
|---|---|---|---|---|
| Safety | 1.00 | 0.21 | 0.06 | 0.06 |
| Speed | 1.00 | 0.64 | 0.91 | 0.44 |
| Wanderlust | 0.99 | 0.50 | 0.43 | 0.94 |
| Wealth | 1.00 | 0.98 | 1.00 | 1.00 |
| Overall | 1.00 | 0.59 | 0.60 | 0.64 |

### B.3 Anchor Path Control

A degenerate single-class prediction can arise either from genuine out-of-distribution behavior or from a fault in the evaluation path itself. To separate these, we run Llama 3.1-8B gameplay through the exact backbone evaluation path used for the other models, holding feature extraction, scaling, and inference identical, and inspect the shape of the resulting predictions rather than their magnitude.

The control passes cleanly. The frozen Longformer produces a full nine-class prediction spread on the anchor data, with all nine classes represented and the most frequent class accounting for only 14% to 19% of episodes, showing no single-class saturation. The frozen BiLSTM recovers motivation at 1.00 across all four classes. Because the path reproduces healthy, non-degenerate behavior on the anchor, the evaluation path is sound end to end, and the Llama 3.3-70B collapse to a single class is a genuine property of how the frozen predictor responds to that backbone's gameplay, not an artifact of the path.

### B.4 Question-Voice Analysis

The predictor reads the agent's questions, so transfer failure can be examined through how each backbone's questions differ from the anchor's. Three independent lenses agree, and they locate the deficit in linguistic register rather than in role-play quality.

At the surface level, question length separates the backbones cleanly (Table B.4). The Llama 3.1-8B anchor asks short questions averaging 14.2 words; Mistral-7B and Qwen2.5-7B are moderately longer, and Llama 3.3-70B is far longer at 40.5 words on average. In embedding space, a probe separating each backbone from the anchor recovers the same ordering: Mistral is the most anchor-like and Llama 3.3-70B is the most distinct, with near-perfect separability.

Table 13: Question-voice distance from the anchor. Mean words per question, and AUC of a probe separating each backbone's questions from the anchor's (BGE embeddings; 0.5 is indistinguishable).

| Backbone | Mean words/question | Probe AUC vs anchor |
|---|---|---|
| Llama 3.1-8B (anchor) | 14.2 | — |
| Mistral-7B | 18.2 | 0.92 |
| Qwen2.5-7B | 22.8 | 0.99 |
| Llama 3.3-70B | 40.5 | 0.99 |

A blind LLM judge confirms the questions carry a backbone-specific signature. Asked to identify the source backbone of a question from four options, the judge reaches 73% accuracy against a 25% chance baseline over 60 items. The same judge, scoring whether each chosen action is coherent with its question and faithful to its stated alignment, finds Llama 3.3-70B among the highest on both, so its distinctiveness is not a drop in role-play quality. These per-backbone coherence and fidelity scores rest on 12 items each and are read directionally, not as ranked point estimates.

Together the three lenses give a single account. Scaling the backbone up moved its question voice further from the anchor's training manifold, not closer, and the frozen predictor saturates to its prior once that distance is large enough. The transfer failure is therefore a property of distributional distance in linguistic register, distinct from the in-distribution inference ceiling, and the two limits are reported separately throughout.

## C  Agent Simulation Protocol

Agents are implemented as stateful LLM wrappers (Llama 3.1-8B) that maintain a rolling conversation history and communicate with the model via a local inference API hosted on the University of Idaho's Research Computing and Data Services cluster. All inference parameters used server defaults; no temperature, top-p, or sampling overrides were applied.

### C.1  System Prompt Structure

Each agent is instantiated with a full-profile system prompt specifying both alignment and motivation. The prompt instructs the agent that it is playing a Dungeons and Dragons character of the specified alignment motivated by the specified motivation, provides the motivation definition, and references the AD&D Player's Handbook (2nd edition) for the alignment definition. The four motivation definitions provided to agents are as follows:

- **Wealth:** "If it has value, you must have it. You have no qualms about risking life and limb in pursuing riches."

- **Safety:** "Your personal Safety is your concern. Items that protect and ensure your safety are of the utmost importance."

- **Wanderlust:** "You want to explore as much as possible. Items that extend your time or allow you to wander further are important to you."

- **Speed:** "Efficiency is key. Items that help reduce turns and make navigation easier are what you want and must have. Speed is efficiency."

The prompt structures each turn as two sequential phases and instructs the agent to avoid revisiting rooms except in extreme cases. Agents are instructed not to mention their alignment or motivation in any justification field. The complete episode objective as presented to the agent is to stay true to the assigned alignment and motivation and find the exit.

## C.2   Turn Structure

Each turn consists of two sequential phases.

In the question phase, the agent receives the room description and available actions and returns a JSON object containing `Question` and `Justification`. The agent is instructed to ask one relevant question about the room or available actions, grounded in its alignment and motivation but without mentioning either directly. Questions asking what the agent should do or how it should feel are explicitly prohibited. Questions must concern the room or available actions only.

In the action phase, the agent receives the room description, the available actions, its own question from the question phase, and the dungeon master's response to that question. It returns a JSON object containing `NumericAnswer`, `Direction`, and `Justification`. Available actions are presented in the format `(#) **Action**`.

## C.3   Conversation History Management

Agents maintain a rolling conversation history capped at 7 turns. History is stored as a list of role/content pairs and prepended to each new prompt. When the cap is exceeded, the oldest turns are dropped. This windowing strategy reflects findings from preliminary testing in which reducing history below three recent turns produced measurable performance degradation.

Question phase requests do not use conversation history; each question prompt is sent fresh against the system prompt only. Action phase requests include the full rolling history, ensuring the agent has access to recent context when selecting actions.

## C.4   Hallucination Handling and Retry Logic

Each action response is validated against the list of valid action indices provided in the prompt. If the agent returns an index not present in that list, the request is retried with corrective feedback appended identifying the invalid selection and listing valid options. Up to three retries are attempted for malformed JSON and invalid action selections, with delays beginning at 1 second and increasing by 0.25 seconds per attempt. Five consecutive hallucinations after retries terminate the episode; these episodes are excluded from all datasets as described in Appendix A.

API-level backend errors trigger unlimited retries on the same backoff schedule.

# D Reproducibility Details

## D.1 Data Preprocessing Pipeline

Raw behavioral sequences undergo the following preprocessing pipeline before model training.

1. **Blocklist exclusion**: Episodes generated during environment development are excluded, leaving 13,153 of 17,400 raw Phase 3 episodes.

2. **Per-model completeness cleaning**: Each model's parquet is flattened and cleaned independently. The Longformer parquet drops Got-Stuck episodes with 8 or fewer sequences and Complete episodes with 3 or fewer sequences; the BiLSTM parquet drops episodes with fewer than 10 sequences. The episode sets are then intersected so both parquets hold the identical 11,495 episodes.

3. **Canonical dataset filters**: The canonical dataset retains the intersection of episodes passing the Longformer filter (timesteps with `active_encounter = True` OR `active_loot = True`; at least 4 active sequences per episode) and the BiLSTM filter (at least 25 total sequences per episode), yielding 10,338 episodes.

4. **Sequence truncation/padding**: Sequences are truncated or zero-padded to 214 timesteps (95th percentile of sequence lengths).

5. **Feature standardization**: Non-embedding positional features are standardized using scikit-learn's `StandardScaler`, fit on training data only.

6. **Canonical splitting**: A single episode-level split (70/15/15 train/validation/test) is computed once, stratified on the 9-class alignment, and materialized to separate per-model parquet files read by every model, including the combined `both`-mode run.

## D.2 Behavioral Consistency Scoring

Each encounter and loot prompt is generated with profile-specific content: keywords, framings, and action options are written to appeal to particular alignment and motivation combinations. When an agent selects an action at an encounter, it receives a score reflecting how closely that action matches the option designed for its assigned profile. Full marks are awarded for selecting the profile-targeted action; partial or zero marks are awarded for other selections, weighted by their distance from the targeted option. Per-encounter scores are summed over a complete episode and divided by the maximum possible score achievable had the agent selected the profile-targeted action at every encounter, yielding a normalized behavioral consistency score in $[0, 1]$. The threshold of 0.7 (inclusive) is enforced when episodes are generated; episodes below it are never uploaded to the corpus.

## D.3 Behavioral Consistency Filtering

Each sequence records `actual_points` and `expected_points`. The behavioral consistency score for a episode is computed as:

$$\text{Coherence} = \frac{\sum \texttt{actual\_points}}{\sum \texttt{expected\_points}} \qquad (2)$$

yielding a normalized score in $[0, 1]$. episodes scoring below 0.7 are excluded prior to any classification experiments, retaining only episodes where the agent behaved in accordance with its assigned profile on at least 70% of available behavioral signal. This threshold additionally mitigates distortion from LLM safety alignment overrides: episodes where the model resists Evil-profile instructions would produce low consistency scores and are removed before classification. The scoring procedure is described in full in Section 3.6 of the main paper.

Post-balancing, the BiLSTM dataset contains between 93 and 129 episodes per profile (mean 112.9); the Longformer dataset contains between 80 and 119 episodes per profile (mean 98.1).

### D.4    Training hyper-parameters

Table D.4 reports the hyper-parameters for the Player2Vec (Longformer) alignment model. The same configuration is used for single-stage training, the primary method, and for the curriculum ablation; stage-specific overrides for the curriculum variant appear in Table F.4 in Appendix F.

Table 14: Training hyper-parameters for the Player2Vec (Longformer) alignment model.

| Parameter | Value | Notes |
| --- | --- | --- |
| *Optimizer* | | |
| Optimizer | AdamW | |
| Base learning rate | $5 \times 10^{-5}$ | Stage-specific overrides apply |
| Weight decay | 0.01 | |
| Gradient clipping | 1.0 | Max gradient norm |
| *Scheduler* | | |
| Scheduler | Cosine | With warm restarts |
| Warmup epochs | 5 | |
| Minimum learning rate | $1 \times 10^{-5}$ | |
| *Regularization* | | |
| Dropout | 0.3 | All dropout layers |
| Label smoothing | 0.1 | Cross-entropy loss |
| Class weighting | Inverse frequency | Balanced classes |
| *Architecture* | | |
| Hidden size | 512 | |
| Attention heads | 8 | |
| Transformer layers | 6 | |
| Attention window | 256 | Local attention span |
| Max sequence length | 214 | 95th percentile |
| *Training* | | |
| Batch size | 16 | |
| Early stopping patience | 10–20 | Stage-dependent |

## E    Architecture Specifications

This appendix provides complete specifications for all model variants evaluated. The primary models are the BiLSTM and the Longformer (Player2Vec) in its selected configuration (lr $1 \times 10^{-5}$, hidden size 256). Additional variants evaluated during Phase 3 include a GRU router-specialist network and several ablation configurations.

### E.1    Feature Encoder

All deep learning models share a common input representation. Each timestep combines 768-dimensional BGE embeddings across six text fields (room description, encounter description, action text, player question, loot description, state transition) with engineered features capturing temporal dynamics, spatial state, and action statistics, yielding a 4,653-dimensional input vector per timestep. The feature encoder projects this to a 512-dimensional representation via a linear layer followed by LayerNorm, ReLU activation, and dropout.

### E.2 BiLSTM

The BiLSTM processes sequences of 512-dimensional encoded features. Architecture details are as follows: 2 bidirectional LSTM layers with 512 hidden units per direction (1,024 effective hidden dimension), yielding 7.7M parameters. Sequence aggregation uses a learned attention mechanism combining attention-weighted sum, max pooling, and mean pooling. The classification head is a 3-layer MLP. Sequences are padded or truncated to 214 timesteps.

Additional BiLSTM variants evaluated during architecture search are reported in Table 21. Increasing model capacity did not improve alignment inference performance; the 5.55M parameter GRU router-specialist variant underperformed the 1.79M parameter Direct-36 BiLSTM variant, ruling out capacity as the binding constraint.

### E.3 GRU Router-Specialist

The GRU router-specialist network implements a two-stage hierarchical classification strategy. A shared `ClusterRouter` processes input sequences through a bidirectional GRU and routes each sequence to one of five alignment clusters: Lawful Aligned (LG, LN, LE; 12 profiles), Chaotic Aligned (CG, CN, CE; 12 profiles), True Neutral (TN; 4 profiles), Neutral Good (NG; 4 profiles), and Neutral Evil (NE; 4 profiles). Sequence aggregation in the router concatenates attention-weighted sum, max pooling, and mean pooling representations before passing through a linear routing head that produces cluster logits.

A bank of five `ClusterSpecialist` modules, one per cluster, each implements an independent bidirectional GRU that predicts the specific profile within its assigned cluster. During training, all specialists receive the input and contribute to the loss. During inference, only the specialist corresponding to the router's predicted cluster is evaluated, and its output is projected to the full 36-class space with non-cluster profiles masked to $-\infty$.

Despite its larger parameter count (5.55M), this architecture achieved 19.8% profile accuracy, below the 1.79M parameter BiLSTM at 25.0%. The result indicates that the hierarchical decomposition does not provide a useful inductive bias for this task: cluster-level routing errors propagate to the specialist stage and cannot be recovered.

### E.4 Player2Vec (Longformer)

Player2Vec adapts Longformer's local attention mechanism for behavioral sequence classification. The architecture comprises three components. The feature encoder (described above) projects the 4,653-dimensional input to 512 dimensions. The Longformer backbone consists of 6 transformer layers with 8-head local attention (window size 256) and absolute positional embeddings. Sequence aggregation uses masked mean pooling: for a sequence of length $L$ with hidden states $\{h_1, \ldots, h_L\}$ and padding mask $m$,

$$h_{\text{pooled}} = \frac{\sum_{i=1}^{L} m_i \cdot h_i}{\sum_{i=1}^{L} m_i} \tag{3}$$

The classification head is a 2-layer MLP ($512 \rightarrow 256 \rightarrow 9$). Total parameters: 21.4M. Maximum sequence length is configurable; experiments used 214 timesteps (95th percentile of sequence lengths). Table 15 provides a direct comparison with the BiLSTM.

### E.5 Motivation BiLSTM (Final Ensemble)

The final ensemble's motivation inference component is a smaller, task-specific BiLSTM trained exclusively for 4-class motivation classification. The model uses 2 bidirectional LSTM layers with 128 hidden units per direction (256 effective hidden dimension), feature projection to 256 dimensions, and the same attention plus max and mean pooling aggregation strategy as the search-phase variants, yielding 1.3M total parameters. Sequences are padded or truncated to 214 timesteps. This model achieves 99.75% validation accuracy and 100% test accuracy on motivation classification; full training specifications are described in Section 3.6..

Table 15: Architecture comparison between alignment inference search-phase models. The final ensemble motivation BiLSTM is described separately below.

| Component | BiLSTM (search) | Player2Vec (Longformer) |
|---|---|---|
| Parameters | 7,701,645 | 21,406,464 |
| Layers | 2 | 6 |
| Hidden size | 512 (1,024 bidir) | 512 |
| Attention type | Scalar + position | Multi-head local (8 heads) |
| Attention scope | Full sequence | Local window (256 tokens) |
| Position encoding | Learnable weights | Absolute positional embeddings |
| Aggregation | Attention + max + mean | Masked mean pooling |
| Classifier | 3 layers | 2 layers |
| Max sequence length | 214 | 214 |

## F  Extended Results

### F.1  Full 36-Class Profile Prediction

The 36-class profile prediction system combines two independently trained models: the BiLSTM for motivation inference and the Longformer for alignment inference. Under the canonical split both models are trained and evaluated on the same held-out episodes, so the joint 36-class accuracy is measured directly from a combined (`both`-mode) run rather than estimated from the two marginal accuracies, providing the empirical joint confusion structure requested in review. Because motivation inference is near-perfect, joint profile accuracy tracks alignment accuracy and profile-level error is dominated by belief-system confusion.

On the shared canonical test set the combined system achieves 34.0% accuracy on full 36-class profile prediction, a 12.2× improvement over the 2.78% (1/36) random baseline. The joint confusion matrix appears in Figure 3.

### F.2  Per-Motivation Accuracy by Alignment

With both models evaluated on the shared canonical test set, per-motivation accuracy can be broken out by alignment directly. Given that BiLSTM motivation accuracy exceeds 98% across all four motivations and all conditions, per-alignment variation is expected to be negligible; the motivation confusion matrix in Table 16 confirms near-diagonal structure with no systematic alignment-correlated errors.

Table 16: Averaged motivation confusion matrix (BiLSTM).

|  | Safety | Speed | Wanderlust | Wealth |
|---|---|---|---|---|
| Safety | 100.0 | 0.0 | 0.0 | 0.0 |
| Speed | 0.0 | 100.0 | 0.0 | 0.0 |
| Wanderlust | 0.0 | 0.3 | 99.5 | 0.3 |
| Wealth | 0.0 | 0.0 | 0.5 | 99.5 |

### F.3  Curriculum Learning Algorithm

Algorithm 1 presents the complete 9-stage curriculum learning procedure. The key design principle is weight transfer between stages: each stage initializes from the best checkpoint of the previous stage, allowing the model to build hierarchical representations progressively. Early stages train on binary opposites (e.g., Lawful Good vs. Chaotic Evil), establishing strong feature representations for maximum-contrast distinctions before introducing intermediate categories. All stages use a single train/validation split; the test partition is reserved for final evaluation.

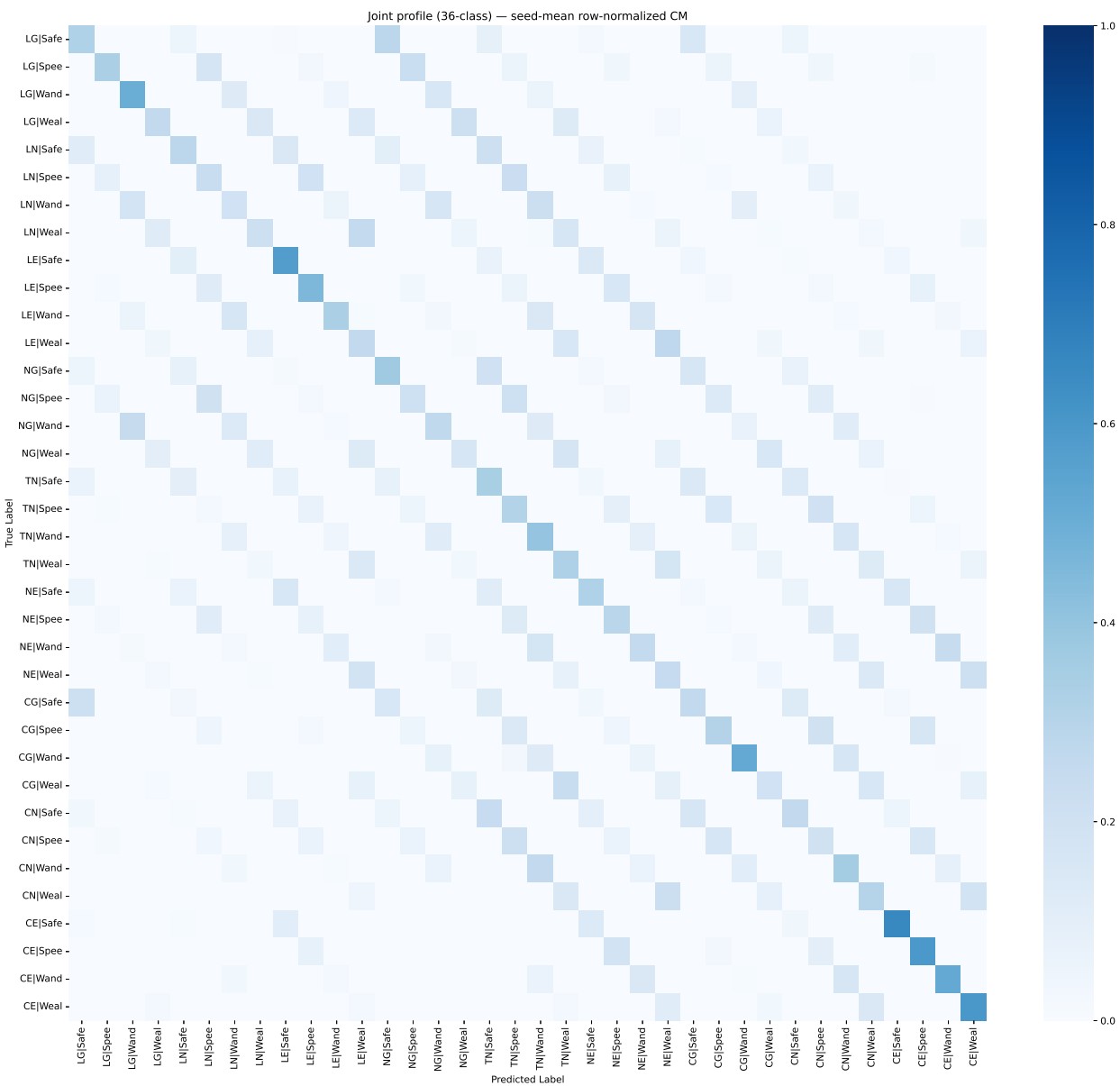

Figure 3: Joint 36-class confusion matrix (seed-mean, row-normalized), ordered by alignment then motivation. Off-diagonal mass forms a block-diagonal stripe: errors preserve motivation while crossing alignment, the inference asymmetry shown directly. Values are omitted at this resolution; Figure 4.4 and Table 16 give the numeric breakdowns.

## F.4    Curriculum Stage Detail

Table F.4 gives the per-stage training schedule for the curriculum variant evaluated in the ablation (Table F.5). Each stage initializes from the best checkpoint of the previous stage. The progressive layer-freezing variant additionally freezes early layers in later stages; as the ablation shows, this freezing degrades performance rather than preserving it, which is why the primary results use single-stage training.

---

**Algorithm 1** 9-Stage Curriculum Learning for Alignment Classification

---

**Require:** Model $M$, Dataset $D$, Stage configurations $\{S_1, \ldots, S_9\}$
**Ensure:** Trained model $M^*$
1: Split $D$ once into $D^{\text{train}}, D^{\text{val}}, D^{\text{test}}$ (episode-level, stratified, fixed seed); hold out $D^{\text{test}}$
2: Initialize model parameters $\theta$
3: checkpoint $\leftarrow$ None
4: **for** stage $s = 1$ to $9$ **do**
5:    $\mathcal{A}_s \leftarrow$ alignments for stage $s$
6:    $D_s^{\text{train}} \leftarrow \{(x, y) \in D^{\text{train}} : y \in \mathcal{A}_s\}$
7:    $D_s^{\text{val}} \leftarrow \{(x, y) \in D^{\text{val}} : y \in \mathcal{A}_s\}$
8:    Initialize optimizer with stage learning rate $\eta_s$
9:    best_acc $\leftarrow 0$, patience_counter $\leftarrow 0$
10:    **for** epoch $= 1$ to max_epochs$_s$ **do**
11:      **for** batch $(X, Y)$ in $D_s^{\text{train}}$ **do**
12:        $\hat{Y} \leftarrow M(X; \theta)$
13:        $\mathcal{L} \leftarrow \text{CrossEntropy}(\hat{Y}, Y)$
14:        $\theta \leftarrow \theta - \eta_s \cdot \text{clip}(\nabla_\theta \mathcal{L}, 1.0)$
15:      **end for**
16:      val_acc $\leftarrow \text{Evaluate}(M, D_s^{\text{val}})$
17:      **if** val_acc $>$ best_acc **then**
18:        best_acc $\leftarrow$ val_acc
19:        checkpoint $\leftarrow \theta$
20:        patience_counter $\leftarrow 0$
21:      **else**
22:        patience_counter $\leftarrow$ patience_counter $+ 1$
23:      **end if**
24:      **if** patience_counter $\geq$ patience$_s$ **then**
25:        **break**
26:      **end if**
27:    **end for**
28:    $\theta \leftarrow$ checkpoint
29: **end for**
30: **return** $M^* \leftarrow M(\cdot; \text{checkpoint})$

---

Table 17: Complete 9-stage curriculum schedule with per-stage training hyper-parameters.

| Stage | Classes | Max Epochs | Patience | LR | Batch |
|-------|---------|------------|----------|----|----|
| 1a | LG vs CE (2) | 30 | 15 | $5 \times 10^{-5}$ | 32 |
| 1b | LE vs CG (2) | 50 | 15 | $5 \times 10^{-5}$ | 32 |
| 1c | LN vs CN (2) | 50 | 15 | $5 \times 10^{-5}$ | 32 |
| 1d | NG vs NE (2) | 70 | 20 | $5 \times 10^{-5}$ | 32 |
| 2a | Corner quad (4) | 100 | 20 | $5 \times 10^{-5}$ | 16 |
| 2b | Edge quad (4) | 100 | 20 | $5 \times 10^{-5}$ | 16 |
| 3 | 8 non-center | 100 | 20 | $5 \times 10^{-5}$ | 16 |
| 4 | All 9 alignments | 100 | 20 | $5 \times 10^{-5}$ | 16 |
| 5 | Full 36 profiles | 200 | 10 | $5 \times 10^{-5}$ | 16 |

## F.5   Baseline Comparisons

Table 19 reports Phase 1 traditional ML baselines using aggregated behavioral features. These establish that learnable signal exists in the data from the outset, and that the random baseline of 11.1% (alignment)

Table 18: Longformer alignment ablation on the canonical split. All variants trained on the canonical training split and evaluated on the held-out test split; selected on best validation accuracy with test accuracy reported post-hoc. Metric is 9-class alignment accuracy.

| Variant | Varies from anchor | Val Acc. | Test Acc. | Macro-F1 |
|---|---|---|---|---|
| Single-stage (anchor) | base config | 30.8% | 29.3% | 0.287 |
| Curriculum, no freeze | training schedule | 28.1% | 28.2% | 0.278 |
| Curriculum, with freeze | schedule + freezing | 11.6% | 11.6% | 0.023 |
| lr $1 \times 10^{-5}$ | learning rate | 31.9% | 31.2% | 0.316 |
| lr $1 \times 10^{-4}$ | learning rate | 11.1% | 11.0% | 0.022 |
| dropout 0.1 | dropout | 29.0% | 27.7% | 0.272 |
| dropout 0.5 | dropout | 30.3% | 30.2% | 0.289 |
| batch 32 | batch size | 30.2% | 31.6% | 0.301 |
| hidden 256 | hidden size | 32.8% | 31.3% | 0.297 |
| layers 4 | transformer layers | 29.7% | 27.3% | 0.246 |

and 25.0% (motivation) is substantially exceeded even by non-sequential methods. The gap between Phase 1 ML baselines and the final Longformer performance isolates the contribution of sequential modeling.

Table 19: Phase 1 baseline performance with aggregated features.

| Method | Alignment (9) | Motivation (4) | Profile (36) |
|---|---|---|---|
| Naive Bayes | 24.4% | 51.2% | 10.39% |
| XGBoost | 24.7% | 48.3% | 9.87% |
| NB Multiclass | 22.1% | 47.6% | 8.92% |
| Random Baseline | 11.1% | 25.0% | 2.78% |

Table 20 traces the incremental architecture improvements leading to the 24% LSTM ceiling. The non-reproducible 30.3% outlier reflects the approximately 10% of runs that collapse to baseline without a fixed random seed.

Table 20: LSTM architecture evolution (Phase 2).

| Configuration | Profile Acc. | Modification |
|---|---|---|
| Basic LSTM | 2.8% | Random-level performance |
| + Multi-pooling | 14.8% | Max + mean pooling |
| + Dimensionality reduction | 21.0% | 512→64 text embeddings |
| + Bidirectional | 22.8% | Forward + backward context |
| + Decomposed heads | 24.2% | Separate alignment/motivation heads |
| Best single run | 30.3%* | *Non-reproducible outlier |
| Stable performance | ∼24–25% | Consistent ceiling |

Table 22 reports Phase 3 signal enhancement results. Despite a 2.7× increase in value-testing encounters, LSTM performance improved only 3.8 percentage points, confirming that the recurrent ceiling is architectural rather than data-limited.

### F.6 Model Selection

All configurations in both sweeps were enumerated before training; the final configuration was selected on seed-mean validation metrics, with validation macro-F1 (Table 25) breaking a tie in validation accuracy

Table 21: Architecture evolution and ablation results (Phase 3, 10×10 grid). Profile accuracy combines alignment and motivation inference.

| Architecture | Parameters | Profile Acc. | Key Feature |
|---|---|---|---|
| Improved LSTM (baseline) | 1.86M | 20.4% | Decomposed heads |
| Direct-36 | 1.79M | 25.0% | End-to-end learning |
| Two-Stage Hierarchical | 2.31M | 20.6% | Cascaded inference |
| GRU Router-Specialist | 5.55M | 19.8% | Router + specialists |
| BiLSTM + BGE | 7.70M | 19.6% | Enhanced embeddings |
| Player2Vec (Longformer) | 21.4M | 21.5% | Transformer attention |

Table 22: Impact of signal enhancement on LSTM performance (Phase 3).

| Configuration | Alignment | Motivation | Profile |
|---|---|---|---|
| Baseline (30% encounters) | 19.2% | 99.7% | 19.2% |
| High density (81% encounters) | 23.0% | 99.9% | 23.0% |
| + Explanatory queries | 23.0% | 99.9% | 23.0% |

between the two leading configurations, and test performance reported post-hoc. The selected configuration, lr $1\times10^{-5}$ + hidden 256, is the primary model for all subsequent analysis.

Table 23: Factorial sweep over single-axis qualifiers and their combinations. Three seeds per variant; final configuration selected on seed-mean validation accuracy, test reported post-hoc.

| Variant | Seed 7 | | Seed 1007 | | Seed 37 | | Seed mean | |
|---|---|---|---|---|---|---|---|---|
| | val/test | F1 | val/test | F1 | val/test | F1 | val/test | F1 |
| anchor | 28.9/27.6 | 0.271 | 27.9/28.9 | 0.277 | 29.3/28.8 | 0.263 | 28.7/28.4 | 0.270 |
| lr $1\times10^{-5}$ | 32.5/33.4 | 0.321 | 30.7/30.5 | 0.296 | 33.1/31.2 | 0.304 | 32.1/31.7 | 0.307 |
| hidden 256 | 31.4/32.1 | 0.316 | 33.2/32.3 | 0.310 | 33.0/32.6 | 0.319 | 32.5/32.3 | 0.315 |
| batch 32 | 31.8/30.7 | 0.298 | 30.7/31.1 | 0.298 | 30.5/30.9 | 0.284 | 31.0/30.9 | 0.293 |
| lr $1\times10^{-5}$ + hidden 256 | 31.4/33.1 | 0.331 | 33.3/34.6 | 0.339 | 32.8/34.3 | 0.344 | 32.5/34.0 | 0.338 |
| lr $1\times10^{-5}$ + batch 32 | 33.1/30.3 | 0.293 | 31.6/30.7 | 0.302 | 31.9/31.8 | 0.316 | 32.2/30.9 | 0.304 |
| hidden 256 + batch 32 | 32.1/31.4 | 0.295 | 33.4/31.8 | 0.293 | 31.8/33.4 | 0.328 | 32.4/32.2 | 0.305 |
| All | 31.3/32.1 | 0.308 | 32.3/31.8 | 0.310 | 31.5/30.5 | 0.295 | 31.7/31.5 | 0.304 |

Table 24: Factorial sweep summary: seed-mean $\pm$ sd over three seeds per configuration, sorted by validation accuracy. Validation macro-F1 (0.323 vs. 0.317) breaks the tie between the two leading configurations; test performance is reported post-hoc.

| Configuration | Val Acc. (%) | Test Acc. (%) | Test Macro-F1 |
|---|---|---|---|
| hidden 256 | $32.5 \pm 1.0$ | $32.4 \pm 0.3$ | 0.315 |
| lr $1\times10^{-5}$ + hidden 256 | $32.5 \pm 1.0$ | $34.0 \pm 0.8$ | 0.338 |
| hidden 256 + batch 32 | $32.4 \pm 0.9$ | $32.2 \pm 1.0$ | 0.305 |
| lr $1\times10^{-5}$ + batch 32 | $32.2 \pm 0.8$ | $30.9 \pm 0.8$ | 0.304 |
| lr $1\times10^{-5}$ | $32.1 \pm 1.2$ | $31.7 \pm 1.5$ | 0.307 |
| lr $1\times10^{-5}$ + hidden 256 + batch 32 | $31.7 \pm 0.5$ | $31.5 \pm 0.8$ | 0.304 |
| batch 32 | $31.0 \pm 0.7$ | $30.9 \pm 0.2$ | 0.293 |
| Anchor (single-stage) | $28.7 \pm 0.7$ | $28.4 \pm 0.7$ | 0.270 |

Table 25: Validation macro-F1 tiebreak between the two configurations tied on seed-mean validation accuracy. F1 computed per seed on the validation split, then averaged; the combined configuration is selected.

| Configuration | n | Val Acc. (%) | Val Macro-F1 |
|---|---|---|---|
| hidden 256 | 3 | $32.54 \pm 0.98$ | $0.3173 \pm 0.0064$ |
| lr $1\times10^{-5}$ + hidden 256 | 3 | $32.51 \pm 0.98$ | $\mathbf{0.3226 \pm 0.0084}$ |

