# OpenReview forum: "Behavioral Inference at Scale: The Fundamental Asymmetry Between Motivations and Belief Systems"
_TMLR — Under review for TMLR_

### Review · Reviewer_XEGB · 2026-05-17

**Summary Of Contributions:**

This paper provides an empirical study of how effective behavioral inference on LLM generated actions (in a grid world / dungeons and dragons-like environment) is, using a second neural network to try to distinguish between actions generated by different profiles. The authors find that while motivations can be easily and almost perfectly identified, alignment (good/neutral/evil, lawful/neutral/chaotic) is harder to identify. A transformer is also better at predicting alignment than a LSTM.

Strengths:
- Ablation studies on the various design choices made (e.g., accuracy through the curriculum stages, impact of adding the MFT features)
- The authors test on a large dataset of behavioral sequences.

Weaknesses:
- I am not sure whether this paper is of interest to the broader TMLR community. It feels more like a behavioral research paper than a machine learning paper to me.
- I am a little unclear on what the different belief systems and alignments would have on the gameplay. E.g., an example of what an evil/chaotic player would do in a particular scenario versus a good/lawful player.
- On a related note, more information in general about the game in the main text would be helpful. The game has a lot more bells and whistles than a “standard” grid world environment, which was confusing for a while.
- I believe the main takeaway is that it is hard to discern alignment based off of LLM behavior, but I am not sure what the implication for this is. Is the LLM a stand-in for human behavior that is easier to collect? If so, is this a useful stand-in? (And is this more a behavioral science paper than a machine learning paper?) If not, what are the implications of the fact that it is difficult to fullly discern between different LLM agents that are prompted to follow different alignments?

**Audience:**

No

**Audience Explanation:**

I think a main weakness of this paper is whether this paper is of interest to the TMLR community. It feels more like a behavioral research paper than a machine learning paper. While I can sort of see the implications for AI safety, I am not an expert in that field. To me, the experiments feel a little contrived if the goal is to say something about AI safety, since the experiments do not seem to test realistic scenarios.

**Broader Impact Concerns:**

No additional concerns.

**Claims And Evidence:**

Yes

**Claims Explanation:**

Yes. My takeaway of the main result is that it is difficult to identify alignment, which is a more nebulous term, than motivations, which perhaps have a more easily understandable mapping to actions. One claim I think is not fully supported is the difference between LSTMs and transformers in being able to identify belief systems, given that the transformers were trained with curriculum learning while the LSTMs were not.

**Requested Changes:**

- A broader discussion on the implications of the results would be helpful. In particular, I think a weak point of the paper is the motivation behind a study like this for the machine learning community.
- I think this paper would benefit from more discussion on the game environment, in the main text. The game environment differs significantly from a “normal” grid world environment seen in reinforcement learning contexts, which made it hard to understand what “good”, “neutral”, “evil”, “greedy”, etc. actions would correspond to. The appendix helps clarify it a bit (more so for the motivations than the alignments), but perhaps an example of what each behavior would look like in this environment would be beneficial.
- On page 6, the authors write “This represents a 2.5× improvement over baseline and demonstrates that the 19-20% ceiling reflects architectural limitations of recurrent models rather than fundamental task difficulty.” However, I am not sure this is well justified because the comparison is between recurrent networks without curriculum learning with transformers with curriculum learning. In particular the line “Without curriculum learning, transformer performance matches LSTM baselines” suggests to me that the curriculum learning is playing a significant role here.
- How does filtering data based on behavioral consistency scores affect the results? I am wondering if the reported accuracy is increased as a result of removing data that might be hard to classify. It would be interesting to have those results as a comparison.
- How many games are filtered out via this procedure?
- Given the overlap in behavior of agents with different alignments, is there a way to quantify how “close” the ~50% accuracy is to ideal?

More minor questions:
- It is unclear to me how the moral foundations theory features are determined.
- To me, the phrase “empirical bounds on behavioral inference” overstates the results, given that this paper explores just one specific setting of behavioral inference (e.g., one particular game, one particular set of alignments and motivations)
- In the appendix, it is mentioned that the LLM agent receives an answer to the question it asks, but it is unclear where those answers come from.

---

> ### Author Response · Authors · 2026-05-20
> **Initial Response to Review**
>
> Thank you for the thoughtful review. We have substantively addressed all points raised and have a revised manuscript in preparation. We are waiting for any additional reviews before uploading the consolidated revision and posting our detailed point-by-point responses to avoid versioning issues across reviewers.

---

> ### Author Response · Authors · 2026-06-25
> **Review Responses**
>
> Thank you very much for the patience while we addressed your comments and requested changes with the rigor they deserved.  I've provided below a response to each applicable item. In the revised version of the paper, I've also highlighted all changes in blue to help isolate where changes happened.
>
> 1. *Broaden the discussion of why the asymmetry matters to the machine learning and safety community.*
>    - **Response:** We have made the machine-learning and safety relevance explicit: because motivation is recoverable while belief alignment is not, behavioral monitoring can certify what an agent optimizes for but not its values, which bears directly on alignment faking and the limits of behavioral oversight. See Introduction, Discussion (Implications for AI Safety), and the Conclusion.
>
> 1. *Add main-text environment detail plus a concrete example of what each alignment and motivation looks like in gameplay.*
>    - **Response:** We have added the Behavioral Examples by Alignment subsection (Methodology), a table of representative encounters with the action taken, the agent’s justification, and the profile score across contrasting profiles. The fuller environment and character-generation detail is specified in the Environment Specification appendix, to which we direct the reviewer.
>
> 1. *The LSTM versus transformer comparison is confounded by curriculum learning.*
>    - **Response:** The transformer ceiling is now established under single-stage direct training, making the comparison with recurrent models like-for-like; curriculum appears only as an ablation that does not improve on single-stage training. This is reflected in Results (Recurrent Architecture Ceiling, Ablation Study) and the Appendix (Model Selection, Baseline Comparisons).
>
> 1. *Quantify how the coherence filter (≥ 0.7) affects accuracy; concern that it removes hard cases and inflates accuracy.*
>    - **Response:** The 0.7 coherence threshold is enforced at data generation as a quality gate, so episodes in which the agent did not enact its assigned profile are never produced. The question we study is whether a profile is inferable from profile-consistent play, so such episodes lie outside its scope by construction rather than being filtered to raise accuracy. See Methodology (Behavioral Consistency Verification) and Appendix (Behavioral Consistency Scoring).
>
> 1. *How many games does the filter remove?*
>    - **Response:** Per-stage episode and sequence counts are reported in the filtering funnel (Methodology, Table 1); full criteria are in Appendix (Data Preprocessing Pipeline).
>
> 1. *Quantify how close the recovered accuracy is to the ideal.*
>    - **Response:** We quantify this with mutual-information recovery: motivation inference extracts 97% of available information, alignment only 16.3%, less than a third of what the label distribution permits. See Discussion (Structural Interpretation of the Asymmetry).
>
> 1. *Clarify how the moral-foundations (MFT) features are determined.*
>    - **Response:** The moral-foundations features were exploratory and showed no marked improvement, so they are not part of the final method. This is stated in Methodology (Feature Representation).
>
> 1. *Where do the answers to agent-asked questions come from?*
>    - **Response:** The agent is prompted to ask a free-form question each turn, but no answering system was deployed during data collection. The response field reserved in the episode protocol for a dungeon-master answer system was not used, contains only a placeholder, and is never a training feature, so the questions reflect the assigned profile applied to the current room state alone. The question and action phases are described in Appendix (Agent Simulation Protocol, Turn Structure).
>
> 1. *The phrase “empirical bounds on behavioral inference” overstates from a single setting.*
>    - **Response:** We have scoped these claims to the evaluated setting and model scales, noting explicitly that larger architectures may extract additional signal, and the new transferability evaluation broadens the evidence base. See Introduction, Results (Transformer Inference), and Discussion.

---

### Review · Reviewer_mVQy · 2026-05-17

**Summary Of Contributions:**

### Summary:
The paper addresses the following question: Given observable actions, can we infer the internal states (beliefs and motivations) that resulted in those actions? The paper studies this question through a controlled experiment over LLM-based agents with 36 behaviour profiles. Since each LLM-based agent is assigned a profile, the ground truth of their moral alignment and motivational drives is available. The paper then uses a text-based dungeon and dragon environment, where the agents are forced to reveal their moral alignment and motivation through actions. The experiment was done in three phases where the number of actions increased going through the phases. By the end of phase 3, there were 1.5 million behaviour sequences. Given the behaviour sequences, the authors then trained different classifiers to map between the behaviour sequence and the profiles. While the paper tested several architectures, BiLSTM, GRU, and Longformer, their final ensemble used Longformer for the alignment and BiLSTM for the motivation. Additionally, the training of the classifiers were done using curriculum learning. The results suggest that while motivation was recoverable from the behaviour sequence with accuracy of 98-100%, the belief recovery plateau at 24% regardless of the model capacity.


### Strengths:
- The paper provides a large scale empirical study on recovering behavioural profile based on action sequences. Around 1.5 million behaviour sequences were used in the experiments, there are also ablations over the MFT configurations with and without curriculum learning.
- The filtering based on behaviour consistency strengthens the results.
- The main question that the paper is studying is clear.

### Weaknesses:
- I think the conclusion made in section 5.1 on the information theoretic framing is too strong. The results suggest that the models tested (Longformer, BiLSTM, GRU) have a performance ceiling of ~48%, that doesn’t mean that this ceiling will exist for every possible model and training regime. Certainly, we can’t make this conclusion based on the experiments done in the paper, and this limitation can be coming from the models tested or the training methodology.
- According to Appendix E.1 the BiLSTM and the longformer weren’t run on the same test data, which is confusing for me why this choice was made.
- The paper only tested Llama3.1-8B as the LLM-based agent, which is not enough to draw generalizable conclusions about LLM-based agents.

**Audience:**

Yes

**Audience Explanation:**

Studying the internal states of LLM-based agents given a sequence of actions would be of interest to many researchers working on LLMs.

**Broader Impact Concerns:**

The paper already discusses the impact concersn of this work as it relates to AI safety and behavioural inference of LLM agents.

**Claims And Evidence:**

No

**Claims Explanation:**

The paper claims are only partially addressed, I have some concerns regarding overclaims in the conclusions that are mentioned in the weaknesess and the requested changes. Mainly, the main claim around asymmetry between motivation and belief inference is only partially addressed.

**Requested Changes:**

- Soften the claims around the information-theortic limits to conclusions regarding the tested architectures/training regime.
- The Longformer and BiLSTM need to run on the same test data. One of the main results of the paper is about 49% combined accuracy. However, appendix E.1 mentions that this number is calculated analytically not empirically which weakens the results.
- The experiments need to be done with at least one more LLM-based agent besides Llama3.1-8B. The current results only suggests the asymmetry with respect to Llama3.1-8B, there is no evidence that this is generalizable to other LLM-based agents. So, I think either the conclusions are softened to be only specific to Llama3.1-8B or at least run one more model and exhibit the same conclusions about the belief and motivation otherwise the results are not generalizable.

---

> ### Author Response · Authors · 2026-05-20
> **Initial Response to Review**
>
> Thank you for the thoughtful review. We have substantively addressed all points raised and have a revised manuscript in preparation. We are waiting for any additional reviews before uploading the consolidated revision and posting our detailed point-by-point responses to avoid versioning issues across reviewers.

---

> ### Author Response · Authors · 2026-06-25
> **Review Responses**
>
> Thank you very much for the patience while we addressed your comments and requested changes with the rigor they deserved.  I've provided below a response to each applicable item. In the revised version of the paper, I've also highlighted all changes in blue to help isolate where changes happened.
>
> 1. *Soften the information-theoretic limit claims to the tested architectures and training regime.*
>    - **Response:** We have scoped the ceiling claims to the tested architectures and training regime and present the mutual-information recovery as what these models extract rather than a universal bound. See Discussion (Structural Interpretation of the Asymmetry)and Results (Transformer Inference)
>
> 1. *Run the Longformer and BiLSTM on the same test data; the combined accuracy was analytic, not empirical.*
>    - **Response:** Both classifiers now train and evaluate on the shared canonical test partition, and the joint 36-class accuracy is measured directly from a combined run (34.0%) rather than from the product of marginals. The measured confusion matrix is in Appendix (Full 36-Class Profile Prediction).
>
> 1. *Run at least one more LLM agent besides Llama 3.1-8B, or scope the conclusions to Llama.*
>    - **Response:** We have completed this evaluation. New gameplay from three additional generators (Qwen2.5-7B and Mistral-7B as scale-matched laterals, and a Llama-3.3-70B step-up) was passed through the frozen classifiers. Both inferred signals degrade off the training generator: alignment falls to near the nine-class chance floor and motivation drops from near-perfect to roughly 0.59 to 0.64. We therefore scope the quantitative findings to Llama-3.1-8B and present the cross-generator result as a brittleness finding for behavioral monitors, while stating the framework-level claims separately. See Results (Transferability Across Agent-Generating LLMs) and Appendix (Transferability Study Details).

---

### Review · Reviewer_cWDG · 2026-05-28

**Summary Of Contributions:**

In this paper, the authors ask how much of an agent's inner state can be inferred from its behavior alone. They simulate agents with Llama 3.1-8B, give each one of 36 profiles built by crossing nine Dungeons & Dragons alignments with four motivations, and generate over 1.5 million behavioral sequences in grid-worlds. The main finding is an asymmetry. Motivations like wealth-seeking or speed are easy to read off behavior, hitting 98 to 100% accuracy. Belief systems are much harder: recurrent models stall around 24% for all sizes, and even a curriculum-trained transformer reaches only 48.9%. The authors then look at where it breaks down. Evil alignments are caught fairly well, between 60-72%, while True Neutral is almost never recovered and Good alignments blur together. To explain this, they point to identifiability problems in inverse reinforcement learning, the philosophical split between negative and positive duties, and the Knobe effect, arguing that harmful acts leave clearer traces than good ones.

**Audience:**

Yes

**Audience Explanation:**

Behavioral monitoring is increasingly proposed as a way to verify whether an AI system is actually aligned, and this paper directly tests how much we can infer about an agent from its actions alone. The finding that goals are easy to read but values are not indicates about alignment faking and the limits of RLHF.

**Broader Impact Concerns:**

The authors discussed this in details.

**Claims And Evidence:**

No

**Claims Explanation:**

1. The abstract claim of 49% accuracy on full 36-class profile classification, and the 17.6× improvement over the random baseline that follows from it, were not properly measured. As Appendix E.1 states, the motivation BiLSTM and the alignment Longformer were trained and evaluated on separate held-out sets, and the combined accuracy was derived analytically by multiplying the two. This assumes motivation and alignment errors are independent, even though both arise from the same behavioral trace, so the true joint accuracy might be lower.


2. The strong Evil-detectability result (60–72%, Table 4) is based on a single backbone (Llama 3.1-8B), leaving the safety-alignment artifact explanation the authors raise in Section 5.5 as an alternative explanation.

**Requested Changes:**

1. The authors compute the 49% and 17.6× figures by multiplying two accuracies measured on disjoint held-out sets, which assumes that motivation and alignment errors are independent. Since both come from the same behavioral trace, the errors may be correlated, in which case the true joint accuracy would be below the product. The authors should run both classifiers on the same games, and then check the measured 36-class confusion matrix, if the independence assumption fails.

2. The authors draw the strong Evil-detectability conclusion from a single backbone, where safety-alignment artifacts remain as an possible explanation. The authors should replicate on at least two more backbones differing in safety-alignment strength, to separate the behavioral account from the artifact.

---

> ### Author Response · Authors · 2026-06-25
> **Review Responses**
>
> Thank you very much for the patience while we addressed your comments and requested changes with the rigor they deserved.  I've provided below a response to each applicable item. In the revised version of the paper, I've also highlighted all changes in blue to help isolate where changes happened.
>
> 1. *The 36-class accuracy was computed by multiplying accuracies from disjoint held-out sets;
> run both classifiers on the same games and inspect the measured confusion matrix.*
>    - **Response:** The joint accuracy is now measured directly on the shared test partition (34.0%) and the measured 36-class confusion matrix is reported, superseding the analytic estimate; because motivation inference is near-perfect, joint error is dominated by alignment confusion. See Appendix (Full 36-Class Profile Prediction).
>
> 1. *The Evil-detectability result rests on a single backbone; replicate on two or more backbones
> differing in safety-alignment strength.*
>    - **Response:** We have completed replication on three additional backbones (Qwen2.5-7B, Mistral-7B, and a Llama-3.3-70B step-up). Two points bear on the artifact concern. First, because games are filtered at the 0.7 coherence threshold during generation, episodes in which safety training caused the model to hedge an Evil profile are not retained, so a detectable Evil signal among retained episodes does not reduce to refusal behavior. Second, on out-of-distribution gameplay the frozen alignment classifier degrades toward a default class, which limits a clean per-alignment comparison across backbones; we therefore present the transfer result as a brittleness finding and retain the safety-alignment-strength disambiguation as a stated limitation, best settled with weakly aligned or base models. See Results (Transferability) and Discussion (Limitations).